# Convergent, functionally independent signaling by mu and delta opioid receptors in hippocampal parvalbumin interneurons

Xinyi Jenny He, Janki Patel, Connor E Weiss, Xiang Ma, Brenda L Bloodgood, Matthew R Banghart*

Division of Biological Sciences, Neurobiology Section, University of California, San Diego, La Jolla, United States

*For correspondence:
mbanghart@ucsd.edu

Competing interest: The authors declare that no competing interests exist.

**Abstract** Functional interactions between G protein-coupled receptors are poised to enhance neuronal sensitivity to neuromodulators and therapeutic drugs. Mu and delta opioid receptors (MORs and DORs) can interact when overexpressed in the same cells, but whether co-expression of endogenous MORs and DORs in neurons leads to functional interactions is unclear. Here, in mice, we show that both MORs and DORs inhibit parvalbumin-expressing basket cells (PV-BCs) in hippocampal CA1 through partially occlusive signaling pathways that terminate on somato-dendritic potassium channels and presynaptic calcium channels. Using photoactivatable opioid neuropeptides, we find that DORs dominate the response to enkephalin in terms of both ligand sensitivity and kinetics, which may be due to relatively low expression levels of MOR. Opioid-activated potassium channels do not show heterologous desensitization, indicating that MORs and DORs signal independently. In a direct test for heteromeric functional interactions, the DOR antagonist TIPP-Psi does not alter the kinetics or potency of either the potassium channel or synaptic responses to photorelease of the MOR agonist [D-Ala$^2$, NMe-Phe$^4$, Gly-ol$^5$]enkephalin (DAMGO). Thus, aside from largely redundant and convergent signaling, MORs and DORs do not functionally interact in PV-BCs in a way that impacts somato-dendritic potassium currents or synaptic transmission. These findings imply that cross-talk between MORs and DORs, either in the form of physical interactions or synergistic intracellular signaling, is not a preordained outcome of co-expression in neurons.

## Editor's evaluation

This study uses novel photoactivatable opioid ligands and neurophysiological recordings in brain slices to investigate the functional interactions between the delta and mu opioid receptors in parvalbumin-expressing hippocampal interneurons. The authors demonstrate that delta and mu opioid receptors modulate potassium channels without causing heterologous desensitization, indicating that these two opioid receptor types signal independently. These findings extend previous studies by establishing the mechanisms of function of mu and delta opioid receptors in forebrain inhibitory interneurons co-expressing these receptors.

## Introduction

G protein-coupled receptors (GPCRs) regulate cellular physiology through a diverse but limited number of intracellular signaling pathways. In neurons, signaling through multiple GPCRs expressed in the same cell can converge on the same molecular effectors (e.g. ion channels) to regulate

neurophysiological properties such as cellular excitability and neurotransmitter release. Although GPCRs that engage the same family of G proteins (Gα$_s$, Gα$_{i/o}$, or Gα$_q$) are poised to functionally interact through convergent biochemical signaling, it is not clear a priori whether such interactions would actually occur. Examples of interactions include functional synergy, when activation of one receptor subtype enhances activity at the other, or reciprocal occlusion, when the receptor subtypes compete for the same pool of effector molecules. Alternatively, GPCRs have been proposed to functionally interact through the formation of receptor heteromers, such that conformational changes due to ligand binding at one receptor shape agonist-driven signaling at the other.

Mu and delta opioid receptors (MORs and DORs) are both Gα$_{i/o}$-coupled GPCRs that are activated by endogenous opioid neuropeptides such as enkephalin to suppress neuronal excitability and synaptic output. MORs are the primary target of widely used opiate analgesics (e.g. morphine, fentanyl) that are plagued by tolerance, high potential for addiction, and a propensity to cause respiratory depression. MORs and DORs have been proposed to functionally interact such that DOR-targeting drugs could reduce the clinical liabilities of MOR-targeting analgesics. For example, either pharmacological suppression or genetic removal of DOR attenuates morphine tolerance (*Abdelhamid et al., 1991*; *Sánchez-Blázquez et al., 1997*; *Zhu et al., 1999*). Furthermore, co-administration of MOR and DOR agonists produces spinal, supraspinal, and peripheral analgesic synergy (*Porreca et al., 1987*; *Schuster et al., 2015*; *Bruce et al., 2019*). In contrast, antagonism of one receptor has been reported to enhance agonist-driven activity at the other receptor in assays using heterologous receptor expression. These observations have been interpreted to support the existence of MOR/DOR heteromers that interact through direct allosteric coupling (*Fujita et al., 2015*; *Cahill and Ong, 2018*). MOR/DOR heteromers have been specifically implicated as potential therapeutic targets for the treatment of pain, as intrathecal co-administration of the DOR-selective antagonist TIPP-Psi with morphine produces stronger analgesia than morphine alone (*Gomes et al., 2004*). Due to the clinical potential of therapeutic approaches that simultaneously engage MORs and DORs, understanding the mechanisms that underlie their potential for functional interactions is of great importance.

Relatively few studies have investigated functional interactions between endogenous MORs and DORs using sensitive measurements of cellular physiology with the single-cell resolution required to implicate cell-autonomous interactions, as opposed to circuit-level effects. In recordings from neurons in the nucleus raphe magnus after upregulation of DORs in response to chronic morphine treatment, MORs and DORs were found to synergistically suppress inhibitory synaptic transmission through a PKA-dependent pathway, but evidence of heteromers was not observed (*Zhang and Pan, 2010*). Also supporting functionally independent signaling, using both electrophysiological and receptor trafficking experiments, a more recent study of spinal dorsal horn neurons that co-express MOR and DOR did not find evidence for co-internalization or co-degradation after intrathecal administration of either the DOR-selective agonist SNC80 or the MOR-selective agonist [D-Ala$^2$, NMe-Phe$^4$, Gly-ol$^5$]enkephalin (DAMGO) (*Wang et al., 2018*). In contrast, recordings from ventral tegmental area neurons suggested MOR/DOR interactions consistent with heteromer formation (*Margolis et al., 2017*). In that study, TIPP-Psi enhanced DAMGO-evoked membrane potential hyperpolarization, and the MOR antagonist CTOP enhanced hyperpolarization evoked by the DOR agonists DPDPE and deltorphin II. However, at least some of the recordings were from dopamine neurons, which have been shown not to express *Oprm1* mRNA (*Galaj et al., 2020*). Thus, in naïve mice, unequivocal evidence for functional interactions between endogenous MORs and DORs in the same neurons, and in particular, for the existence of MOR/DOR heteromers that impact neuronal physiology, is lacking.

In some brain regions, including the hippocampus, MORs and DORs are established to be co-expressed in the same neurons, such that the receptors and their downstream intracellular signaling pathways are poised to interact (*Chieng et al., 2006*; *Erbs et al., 2015*). In the hippocampus, activation of MORs in GABA neurons contributes to stress-induced memory deficits (*Shi et al., 2020*), whereas DORs may contribute to spatial contextual cue-related memory retrieval (*Le Merrer et al., 2011*; *Le Merrer et al., 2012*; *Le Merrer et al., 2013*). Recently, we reported that MORs and DORs both contribute to opioid-mediated suppression of perisomatic inhibition in the CA1 region of hippocampus, consistent with previous studies of MOR and DOR modulation of synaptic transmission (*Glickfeld et al., 2008*; *Piskorowski and Chevaleyre, 2013*; *Banghart et al., 2018*). In fact, MORs and DORs are well established to regulate inhibitory synaptic transmission in CA1 (*Zieglgänsberger et al., 1979*; *Nicoll et al., 1980*; *Lupica and Dunwiddie, 1991*; *Lupica et al., 1992*; *Lupica, 1995*;

*Svoboda and Lupica, 1998*; *Svoboda et al., 1999*; *Rezaï et al., 2012*). Although a substantial body of work indicates co-expression of MOR and DOR in CA1 parvalbumin basket cells (PV-BCs), which are a primary source of perisomatic inhibition (*Stumm et al., 2004*; *Erbs et al., 2012*; *Faget et al., 2012*; *Yao et al., 2021*), a direct comparison of their neurophysiological actions has not been conducted.

In this study, we explored potential interactions between MORs and DORs in CA1 PV-BCs using recordings from hippocampal slices. In order to obtain precise and sensitive measures of receptor function, we optically probed native MORs and DORs using photoactivatable (caged) opioid neuropeptides (*Banghart and Sabatini, 2012*; *Banghart et al., 2018*). Using this approach, we found that MORs and DORs activate partially overlapping pools of somato-dendritic potassium channels in PV-BCs, and suppress synaptic output from PV-BCs in a mutually occlusive manner. Despite their co-expression and functional redundancy, we did not find evidence of synergy or for heteromers, indicating that MOR and DOR signal in a parallel, functionally independent manner in PV-BCs.

## Results

### Occlusive suppression of hippocampal perisomatic inhibition by MORs and DORs

We first confirmed that both MORs and DORs are co-expressed in PV-BCs using fluorescence in situ hybridization, which revealed that 78% (171/218) of *Pvalb* mRNA-containing neurons with cell bodies

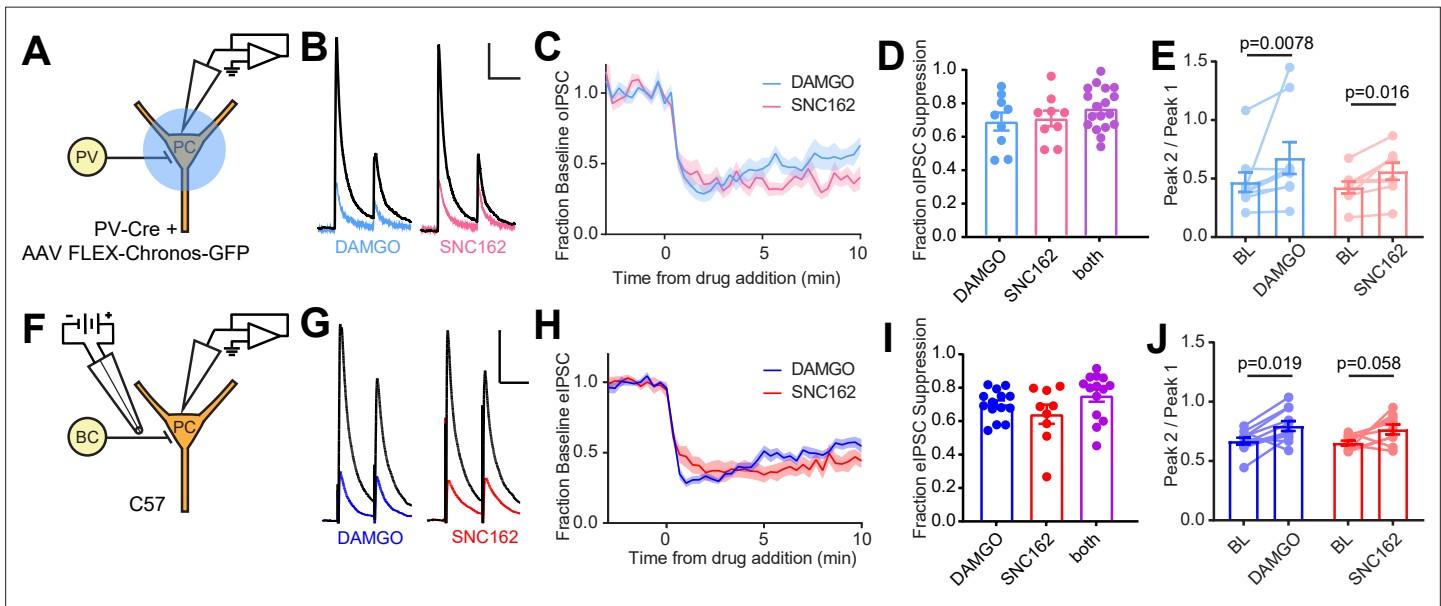

**Figure 1.** Electrophysiological recordings of opioid-sensitive synaptic output from hippocampal parvalbumin basket cells. (**A**) Schematic of the experimental configuration for recording optogenetically evoked inhibitory synaptic transmission in PV-Cre mice. (**B**) Representative optically evoked IPSC (oIPSC) pairs (50 ms interval) recorded from a pyramidal cell. Black traces are the average of six baseline sweeps, and colored traces are the average of six sweeps after addition of either [D-Ala$^2$, NMe-Phe$^4$, Gly-ol$^5$]enkephalin (DAMGO) (1 µM, blue) or SNC162 (1 µM, red). Scale bars: x = 40 ms, y = 100 pA. (**C**) Baseline-normalized, average oIPSC amplitude over time during bath application of DAMGO (n = 9 cells from six mice) or SNC162 (n = 9 cells from seven mice). (**D**) Summary data of double flow-in experiments, comparing oIPSC suppression by DAMGO or SNC162 alone, followed by the other drug. (**E**) oIPSC paired-pulse ratios (Peak 2/Peak 1), before (baseline, BL) and after drug addition. (**F**) Schematic of the experimental configuration for recording electrically evoked inhibitory synaptic transmission in wild-type mice. (**G**) Representative electrically evoked IPSC (eIPSC) pairs (50 ms interval) recorded from a pyramidal cell (as in **B**). Scale bars: x = 40 ms, y = 200 pA. (**H**) Baseline-normalized, average eIPSC amplitude over time during bath application of DAMGO (n = 15 cells from 13 mice) or SNC162 (n = 9 cells from five mice). (**I**) Summary data of double flow-in experiments with electrical stimulation (as in **D**). (**J**) eIPSC paired-pulse ratios (Peak 2/Peak 1), before and after drug addition.

The online version of this article includes the following source data and figure supplement(s) for figure 1:

**Source data 1.** IPSC suppression, paired pulse ratios, and time courses for DAMGO and SNC162.

**Figure supplement 1.** Opioid receptor mRNA in CA1 parvalbumin interneurons and characterization of the neuromodulator sensitivity of CA1 basket cell synaptic output.

**Figure supplement 1—source data 1.** IPSC suppression and time courses for WIN55,212, DAMGO, and SNC162 using opto and estim.

in and around stratum pyramidale contain both *Oprm1* and *Oprd1* mRNA (*Figure 1—figure supplement 1A, B*). To determine if both MORs and DORs are functional in PV-BCs, we virally expressed the light-gated cation channel Chronos in a Cre recombinase-dependent manner in the CA1 region of *Pvalb^Cre* mice and measured the effects of the selective MOR and DOR agonists DAMGO and SNC162, respectively, on light-evoked synaptic transmission using electrophysiological recordings from pyramidal cells (PCs) in acute hippocampal slices (*Klapoetke et al., 2014*). We chose SNC162 due to its exceptional selectivity for DOR over MOR (*Knapp et al., 1996*). To maximize the relative contribution of perisomatic inhibition from PV basket cells, as opposed to dendrite-targeting PV bistratified cells, we restricted the area of illumination to a small region of stratum pyramidale around the recorded PC (*Figure 1A*). Bath perfusion of either DAMGO (1 µM) or SNC162 (1 µM) strongly reduced the optically evoked IPSC (oIPSC) to a similar degree (*Figure 1B–D*). Sequential drug application only slightly increased the degree of suppression compared to either drug alone (DAMGO: 0.69 ± 0.05, n = 9 cells; SNC162: 0.70 ± 0.05, n = 9 cells; both: 0.76 ± 0.03, n = 18 cells; no significant differences, ordinary one-way ANOVA) (*Figure 1D*, *Figure 1—figure supplement 1F*). In both cases, application of pairs of optical stimuli (50 ms apart) revealed small increases in the paired-pulse ratio (PPR) in the presence of the opioid agonist, consistent with a presynaptic mechanism of action for the opioid receptor (BL: 0.47 ± 0.08; DAMGO: 0.68 ± 0.14; n = 9 pairs; p = 0.0078, Wilcoxon matched-pairs signed rank test; BL: 0.42 ± 0.05; SNC162: 0.56 ± 0.07; n = 8 pairs; p = 0.016, Wilcoxon matched-pairs signed rank test) (*Figure 1E*). With sustained application, both the effects of DAMGO and SNC162 appeared to desensitize slightly, with DAMGO showing greater desensitization (*Figure 1—figure supplement 1H, I*) (DAMGO$_{early}$: 0.69 ± 0.05; DAMGO$_{late}$: 0.44 ± 0.07; n = 9 pairs; p = 0.0038, paired t-test; SNC162$_{early}$: 0.70 ± 0.05; SNC162$_{late}$: 0.61 ± 0.06; n = 9 pairs; p = 0.048, paired t-test). These results reveal that both MORs and DORs suppress the output of PV-BCs in a mutually occlusive manner.

To avoid complications due to optical cross-talk between optogenetic tools and photoactivatable peptides in subsequent experiments, we established an electrical stimulation protocol for preferential activation of PV-BC terminals by placing a small bipolar stimulating electrode in stratum pyramidale immediately adjacent to the recorded PC (*Figure 1F*). Recordings were made from PCs near stratum oriens, as these have been shown to receive BC input that is biased toward PV-BCs, as opposed to CCK-BCs (*Lee et al., 2014*). Whereas fast-spiking, presumably PV-BCs have been shown to be opioid, but not cannabinoid sensitive, output from regular-spiking CCK-BCs is suppressed by CB1R, but not MOR activation (*Glickfeld et al., 2008*). Consistent with only a minor contribution to the electrically evoked IPSC (eIPSC) from CB1R-expressing CCK-BCs, bath application of the CB1R agonist WIN55,212 (1 µM) resulted in only modest eIPSC suppression (0.25 ± 0.07, n = 8 cells), and application of WIN55,212 in the presence of DAMGO produced only slightly more suppression than DAMGO alone, although this effect was not significant, suggesting some occlusion (DAMGO: 0.67 ± 0.02, n = 12 cells; WIN55,212 + DAMGO: 0.79 ± 0.03, n = 8 cells; p = 0.14, ordinary one-way ANOVA with Tukey's multiple comparison test) (*Figure 1—figure supplement 1C-E*). Under these electrical stimulation conditions, DAMGO and SNC162 again suppressed the eIPSC to a similar degree, with DAMGO, but not SNC162, producing slight desensitization (*Figure 1—figure supplement 1H*) (DAMGO$_{early}$: 0.70 ± 0.03; DAMGO$_{late}$: 0.41 ± 0.05; n = 13 pairs; p < 0.0001, paired t-test; SNC162$_{early}$: 0.63 ± 0.06; SNC162$_{late}$: 0.57 ± 0.05; n = 9 pairs; p = 0.10, paired t-test). For both eIPSCs and oIPSCs, DAMGO resulted in more desensitization than SNC162 (*Figure 1—figure supplement 1I*) (eIPSC DAMGO: 0.28 ± 0.04; oIPSC DAMGO: 0.25 ± 0.06; eIPSC SNC162: 0.07 ± 0.04; oIPSC SNC162: 0.09 ± 0.04; Skillings-Mack non-parametric test for grouped data (*Mack and Skillings, 1980*), p < 0.0001 for column effects (DAMGO vs. SNC162), p = 0.13 for row effects (eIPSC vs. oIPSC)). As with optogenetic stimulation, DAMGO and SNC162 exhibited strong mutual occlusion of the eIPSC (DAMGO: 0.69 ± 0.02, n = 14 cells; SNC162: 0.63 ± 0.06, n = 9 cells; both: 0.75 ± 0.04, n = 14 cells; no significant differences, ordinary one-way ANOVA), and a small increase in PPR was produced by DAMGO but not SNC162 (BL: 0.67 ± 0.03; DAMGO: 0.80 ± 0.04; n = 11 pairs; p = 0.019, Wilcoxon matched-pairs signed rank test; BL: 0.65 ± 0.02; SNC162: 0.77 ± 0.04; n = 9 pairs; p = 0.055, Wilcoxon matched-pairs signed rank test) (*Figure 1F–J*, *Figure 1—figure supplement 1F*). Although it is possible that an opioid-sensitive population of non-PV interneurons contributes to the opioid-sensitive component of the eIPSC, the effects of DAMGO and SNC162 on the eIPSC and oIPSC were indistinct (no significant difference, two-way ANOVA) (*Figure 1—figure supplement 1G*).

MOR and DOR are thought to exhibit similar affinity for enkephalin, but how this translates to ligand efficacy at native receptors in neurons is not clear. In addition, receptor signaling kinetics could prove to be a sensitive means of detecting functional interactions. To compare the ligand sensitivity and receptor signaling kinetics of MORs and DORs, we turned to photoactivatable derivatives of the MOR and DOR agonist [Leu⁵]-enkephalin (LE) (*Figure 2A*, top) (*Banghart and Sabatini, 2012*). For quantitative pharmacology, we chose to use *N*-MNVOC-LE, which is highly inactive at both DOR and MOR (*Banghart et al., 2018*). In the presence of *N*-MNVOC-LE (6 µM), which is optimized for simultaneous activation of MORs and DORs, application of a strong 5 ms UV light flash 2 s prior to an eIPSC produced a rapid, transient suppression of the eIPSC that recovered within 1–2 min (*Figure 2A and B*). Varying UV light intensity in a graded fashion allowed us to rapidly obtain power-response curves within a single recording. To assess the potency of LE at MORs and DORs, and the relative contributions of the receptors to the eIPSC suppression by LE, we recorded power-response curves in the absence and presence of the MOR- and DOR-selective antagonists CTOP (1 µM) and TIPP-Psi (1 µM), respectively (*Figure 2C*). We chose CTOP over its analog CTAP due to its higher selectivity for MORs. Whereas LE uncaging at the highest light power (84 mW) in the absence of opioid antagonists suppressed synaptic transmission by 63% ± 4%, activation of MORs or DORs alone, which were isolated by antagonizing with TIPP-Psi or CTOP, respectively, suppressed synaptic output by ~40% each. Although the extent of suppression achieved with caged LE was somewhat less than with bath application (*Figure 1I*), the relative contributions of MORs and DORs were similar in both experiments and consistent with mutual occlusion. The power-response curve revealed that LE exhibits approximately threefold greater potency for DORs than MORs in regulating perisomatic inhibition (EC$_{50}$ values in the absence [black, 3.28 ± 0.47 mW] and presence of either CTOP [red, 2.29 ± 0.61 mW] or TIPP-Psi [blue, 9.30 ± 1.40 mW]). Moreover, DOR activation largely accounts for the actions of LE in the absence of antagonists. This could reflect greater affinity for DORs, or more efficacious signaling by DORs than MORs (*Figure 2D*).

We evaluated receptor signaling kinetics using the photoactivatable LE derivative CYLE, which photolyzes within tens of microseconds, such that receptor activation is rate-limiting (*Banghart and Sabatini, 2012*; *Banghart et al., 2018*). In order to sample synaptic transmission at frequencies sufficient to resolve receptor signaling kinetics, we drove eIPSCs in 5 s bouts at 10, 20, and 50 Hz, and photolyzed CYLE (6 µM) after synaptic depression had stabilized to a steady state (*Figure 2E*). To obtain the time constants of synaptic suppression for each receptor, we repeated this experiment in the presence of the selective antagonists and fit the post-flash eIPSC amplitudes with a single exponential function (*Figure 2F*). The time constants we obtained for each pharmacological condition were similar for all three stimulus frequencies (*Figure 2G*). At 20 Hz, DOR (CTOP at 20 Hz, tau = 419 ± 105 ms, n = 11 cells) exhibited kinetics indistinct from the drug-free condition (artificial cerebrospinal fluid [ACSF] at 20 Hz, tau = 259 ± 30 ms, n = 8 cells), but the time constant of MOR-mediated suppression was surprisingly slow (TIPP-Psi at 20 Hz, tau = 683 ± 36 ms, n = 6 cells; p = 0.0046, Kruskal-Wallis test with Dunn's multiple comparisons). At other frequencies, although the MOR kinetics trended toward slower time constants, statistical significance was not observed. We also observed that the extent of eIPSC suppression correlated inversely with the frequency of synaptic stimulation, and that this was most pronounced in the absence of antagonists (*Figure 2H*).

Together, these results suggest that MOR and DOR suppress output from overlapping populations of PV-BC presynaptic terminals, and that this suppression is dominated by DOR, both in terms of sensitivity to LE and response kinetics.

## MORs and DORs suppress GABA release by inhibiting voltage-sensitive Ca²⁺ channels

At least two mechanisms of presynaptic inhibition by Gα$_{i/o}$-coupled GPCRs have been established, but the pathways engaged by opioid receptors in PV-BCs are not known. One potential mechanism involves the inhibition of voltage-sensitive calcium channels (VSCCs) by Gβγ proteins (*Bean, 1989*), whereas the other involves direct suppression of SNARE proteins by Gβγ binding to the C-terminus of SNAP25 (*Blackmer et al., 2001*; *Gerachshenko et al., 2005*; *Zurawski, 2019*; *Hamm and Alford, 2019*). The observed frequency-dependent synaptic suppression is consistent with both mechanisms, as Gβγ binding to VSCCs is reversed by strong depolarization, and elevated Ca²⁺ facilitates

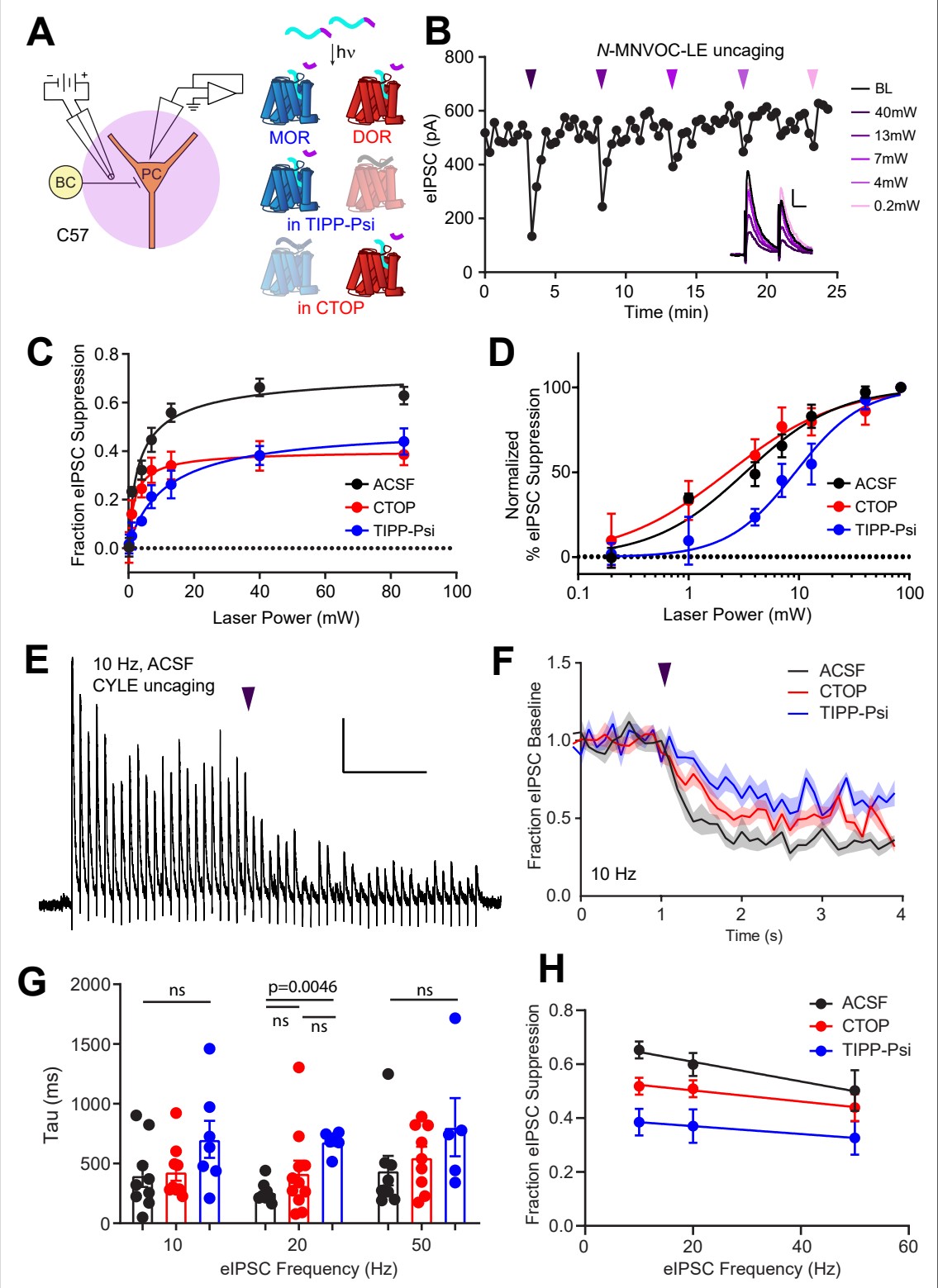

**Figure 2.** Characterization of the potency and kinetics of synaptic modulation by [Leu$^5$]-enkephalin (LE) at mu (MOR) and delta opioid receptors (DOR) using caged peptides. (**A**) Left: Schematic of the experimental configuration for photo-uncaging of opioid neuropeptides while recording electrically evoked inhibitory synaptic transmission in wild-type mice. Right: Schematic of photoreleasing LE (cyan) from *N*-MNVOC-LE or CYLE (cyan with purple caging group) in the presence of selective antagonists to isolate its action on either MOR (blue, in TIPP-Psi) or DOR (red, in CTOP). (**B**) Example recording showing graded suppression of inhibitory synaptic transmission by uncaging *N*-MNVOC-LE at various light intensities. Inset: Example

*Figure 2 continued on next page*

*Figure 2 continued*

electrically evoked IPSC (eIPSCs) before (black) and after LE uncaging at each light intensity. Scale bars: x = 20 ms, y = 100 pA. (C) Linear optical power-response curves of eIPSC suppression as a function of light intensity, in the absence (black, n = 6–12 cells per laser intensity) and presence of either CTOP (red, n = 5–8 cells) or TIPP-Psi (blue, n = 4–10 cells). (D) Logarithmic optical power-response curves of the data in (C) normalized to the maximal eIPSC suppression observed in each condition. (E) Representative recording from a pyramidal cell demonstrating rapid suppression of eIPSC amplitude in response to photoactivation of CYLE during 10 Hz trains of electrical stimuli. Purple arrow represents CYLE uncaging at 2 s into the 10 Hz train. Outward stimulus artifacts are removed for clarity. Scale bars: x = 1 s, y = 100 pA. (F) Average, baseline subtracted and baseline-normalized eIPSC amplitude showing the kinetics of synaptic suppression with electrical stimulation at 10 Hz in the absence (artificial cerebrospinal fluid [ACSF], n = 9 cells from six mice) and presence of either CTOP (n = 12 cells from seven mice) or TIPP-Psi (n = 8 cells from six mice). (G) Time constants of synaptic suppression in response to CYLE photoactivation with an 84 mW light flash at the indicated frequencies of synaptic stimulation. At 20 Hz, the time constant in TIPP-Psi was significantly greater than the time constant without any antagonists. (H) Plot of eIPSC suppression as a function of synaptic stimulation frequency.

The online version of this article includes the following source data for figure 2:

**Source data 1.** Power-response curves and onset kinetics at presynaptic MOR and DOR.

displacement of Gβγ from the SNARE complex by $Ca^{2+}$-bound synaptotagmin (*Park and Dunlap, 1998*; *Brody and Yue, 2000*; *Yoon et al., 2007*).

To ask if MOR and DOR inhibit presynaptic VSCCs in PV-BCs, we imaged action potential (AP)-induced $Ca^{2+}$ transients in presynaptic boutons of PV-BCs using two-photon laser scanning microscopy. PV-BCs were targeted for whole-cell current clamp recordings in *Pvalb$^{Cre}$/Rosa26-lsl-tdTomato* (Ai14) mice with the small molecule $Ca^{2+}$ indicator Fluo5F included in the recording pipette (*Figure 3A*). Line scans across putative boutons were obtained while triggering either one or five APs, before and after bath application of DAMGO, SNC162, or both drugs together (*Figure 3B*).

Individually, DAMGO and SNC162 both caused an ~30% reduction in the peak $\Delta G/G_{sat}$ evoked by either stimulation protocol (DAMGO 27.27% for one AP, 17.73% for five APs, SNC162 31.18% for one AP, 26.55% for five APs). When DAMGO and SNC162 were applied together, these presynaptic $Ca^{2+}$ transients were suppressed by ~40%, on average (DAMGO then SNC162 40.95% for one AP, 38.92% for five APs, SNC162 then DAMGO 46.08% for one AP, 40.85% for five APs) (*Figure 3C and D*). Under the conditions employed, peak $\Delta G/G_{sat}$ is linearly correlated with $Ca^{2+}$ concentration (*Higley and Sabatini, 2008*). Given the nonlinear $Ca^{2+}$ dependence of vesicular fusion, a 30% reduction in presynaptic $Ca^{2+}$ is consistent with the strong suppression of PV-BC IPSCs by MORs and DORs (*Wu and Saggau, 1997*). These results indicate that the inhibition of VSCCs by both MORs and DORs is the most likely mechanism accounting for their effects on inhibitory transmission. Furthermore, the marginal effect of adding a second drug suggests convergence on the same pool of VSCCs.

## Enkephalin generates large outward somato-dendritic currents in PV-BCs primarily through DORs rather than MORs

$G\alpha_{i/o}$-coupled GPCRs, including both MORs and DORs, often hyperpolarize neurons by activating G protein-coupled inward rectifier $K^+$ (GIRK) channels, as well as voltage-gated $K^+$ channels, or by suppressing hyperpolarization-gated cyclic nucleotide (HCN) channels (*Williams et al., 1982*; *North et al., 1987*; *Wimpey and Chavkin, 1991*; *Svoboda and Lupica, 1998*). Although MORs were previously reported to activate outward currents in the somato-dendritic compartment of fast-spiking CA1 BCs, the role of DORs has not been explored (*Glickfeld et al., 2008*). To address this, we performed voltage clamp recordings of opioid-evoked currents in tdTom-labeled cells in *Pvalb$^{Cre}$/Rosa26-lsl-tdTomato* mice (*Figure 4A*). At a holding potential of –55 mV, *N*-MNVOC-LE photoactivation using strong (84 mW) light flashes applied to the soma and proximal dendrites of the recorded neuron evoked rapidly rising outward currents that decayed over ~1 min, similar to previous observations in locus coeruleus (*Figure 4B and C*; *Banghart and Sabatini, 2012*). Surprisingly, blocking MORs with CTOP had no measurable effect on the light-evoked current (ACSF: 81.7 ± 9.6 pA, n = 9 cells; CTOP: 82.5 ± 12.8 pA, n = 10 cells; not significant). In contrast, blocking DOR with TIPP-Psi greatly reduced the current amplitude (TIPP-Psi: 26.4 ± 4.8 pA, n = 11 cells; p = 0.016), and addition of both drugs completely abolished it (CTOP + TIPP-Psi: 7.1 ± 0.09 pA, n = 5 cells; p = 0.0009; Kruskal-Wallis test with Dunn's multiple comparisons). Power-response curves in the presence of each antagonist revealed a larger DOR-mediated than MOR-mediated current (*Figure 4D*). Similar to our observations with presynaptic receptors, LE exhibited greater potency at DORs than MORs in generating outward

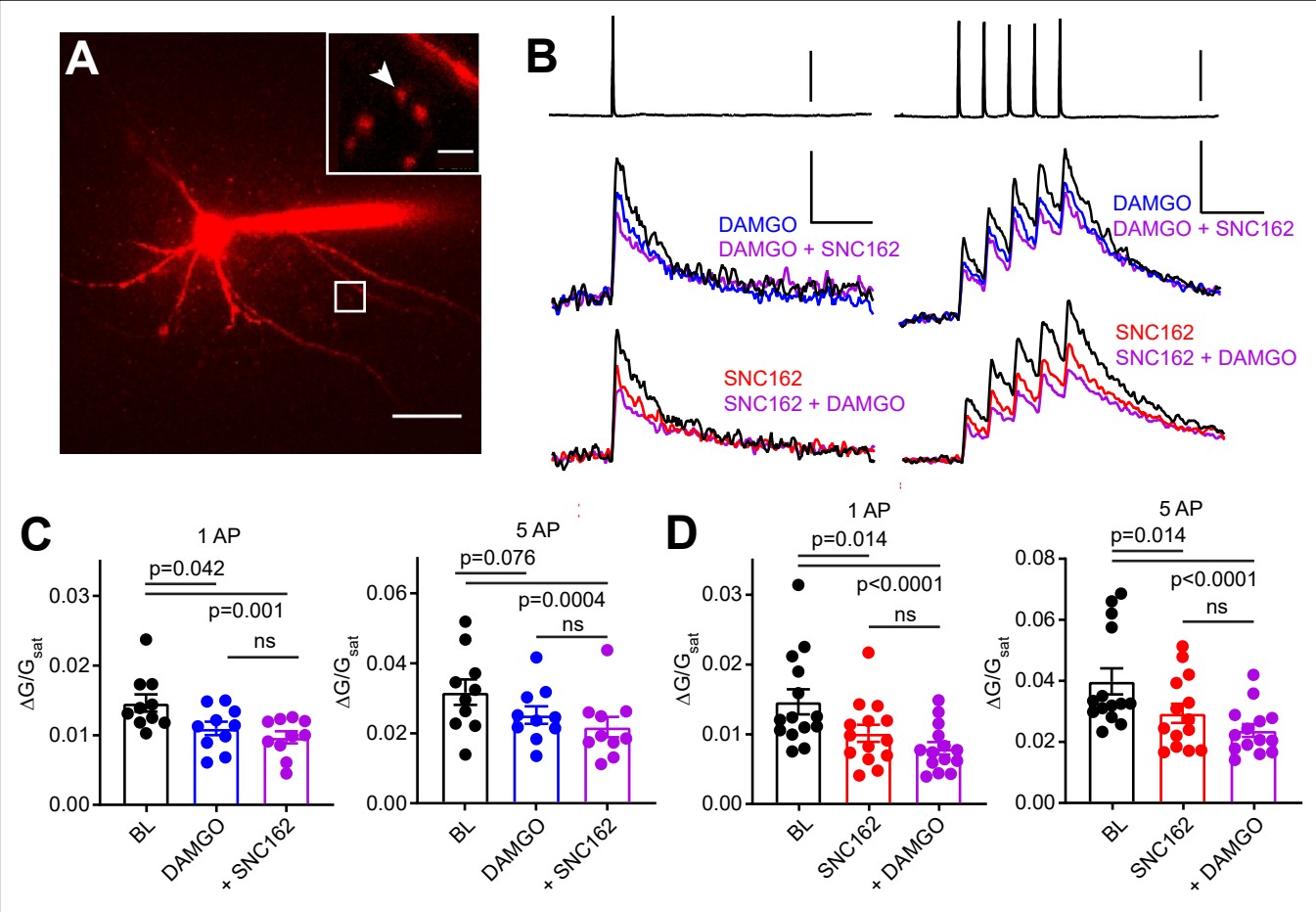

**Figure 3.** Axonal calcium imaging reveals that both mu and delta opioid receptors suppress presynaptic voltage-sensitive calcium channels. (**A**) Two-photon image of a tdTomato-expressing basket cell filled with 30 μM Alexa 594 and 300 μM Fluo-5F in a brain slice taken from a PV-Cre; tdTom mouse. Scale bar: 50 μm. Inset shows the two axonal boutons where the line scan was carried out, with the orientation of the line scan indicated by the arrow. Scale bar: 5 μm. (**B**) Example of either a single action potential (AP) (left) or five APs (right) triggered in the cell body (top), and the resulting averaged, presynaptic $Ca^{2+}$ transients, before and after application of DAMGO (top, blue, n = 8 cells, 16 boutons), SNC162 (red bottom, n = 7 cells, 14 boutons), and both drugs (top and bottom, purple). The transients are measured as the change in green signal ($\Delta G$) , divided by G in saturating $Ca^{2+}$ conditions ($G_{sat}$). Scale bars: top, 50 mV; bottom, x = 100 ms, y = 0.01 (left) or 0.02 (right) ($\Delta G/G_{sat}$. (**C**) Summary of peak $Ca^{2+}$ transients for DAMGO application in response to one AP (left) or five APs (right). One AP: BL 0.014 ± 0.001; DAMGO 0.011 ± 0.001; DAMGO+ SNC162 0.010 ± 0.001 (p = 0.042 and p = 0.0001, n = 10 pairs, Friedman test with Dunn's multiple comparisons) five AP: BL 0.032 ± 0.004; DAMGO 0.025 ± 0.002, DAMGO+ SNC162 0.022 ± 0.003 (p = 0.076 and p = 0.0004, n = 10 pairs). (**D**) Summary of peak $Ca^{2+}$ transients for SNC162 application in response to one AP (left) or five APs (right). One AP: BL 0.014 ± 0.002; SNC162 0.010 ± 0.002; SNC162+ DAMGO 0.008 ± 0.001 (p = 0.014 and p < 0.0001, n = 14 pairs, Friedman test with Dunn's multiple comparisons). Five AP: BL 0.039 ± 0.004; SNC162 0.029 ± 0.003; SNC162+ DAMGO 0.023 ± 0.002 (p = 0.014 and p < 0.0001, n = 14 pairs).

The online version of this article includes the following source data for figure 3:

**Source data 1.** $Ca^{2+}$ transient peaks with and without DAMGO and SNC162.

currents ($EC_{50}$ values of ACSF: 17.55 ± 2.98 mW, CTOP: 7.59 ± 1.26 mW, TIPP-Psi: 28.03 ± 7.14 mW) (*Figure 4E*). Assessment of current activation kinetics with CYLE (6 μM) revealed that, whereas DOR-mediated currents activated with kinetics similar to the MOR currents previously observed in LC neurons, somato-dendritic MOR currents in CA1 PV-BCs activated threefold more slowly, similar to the rate observed for presynaptic MOR in these neurons (ACSF: 275.9 ± 35.7 ms, n = 11 cells; CTOP: 395.3 ± 109.6 ms, n = 6 cells; TIPP-Psi: 844.1 ± 105.2 ms, n = 9 cells; p = 0.019, Kruskal-Wallis test with Dunn's multiple comparisons) (*Figure 4F and G*; *Ingram et al., 1997*; *Banghart and Sabatini, 2012*). The small MOR-mediated currents, coupled with similarly slow signaling kinetics in both the presynaptic and somato-dendritic compartments, suggest that MOR signaling is relatively inefficient in CA1 PV-BCs.

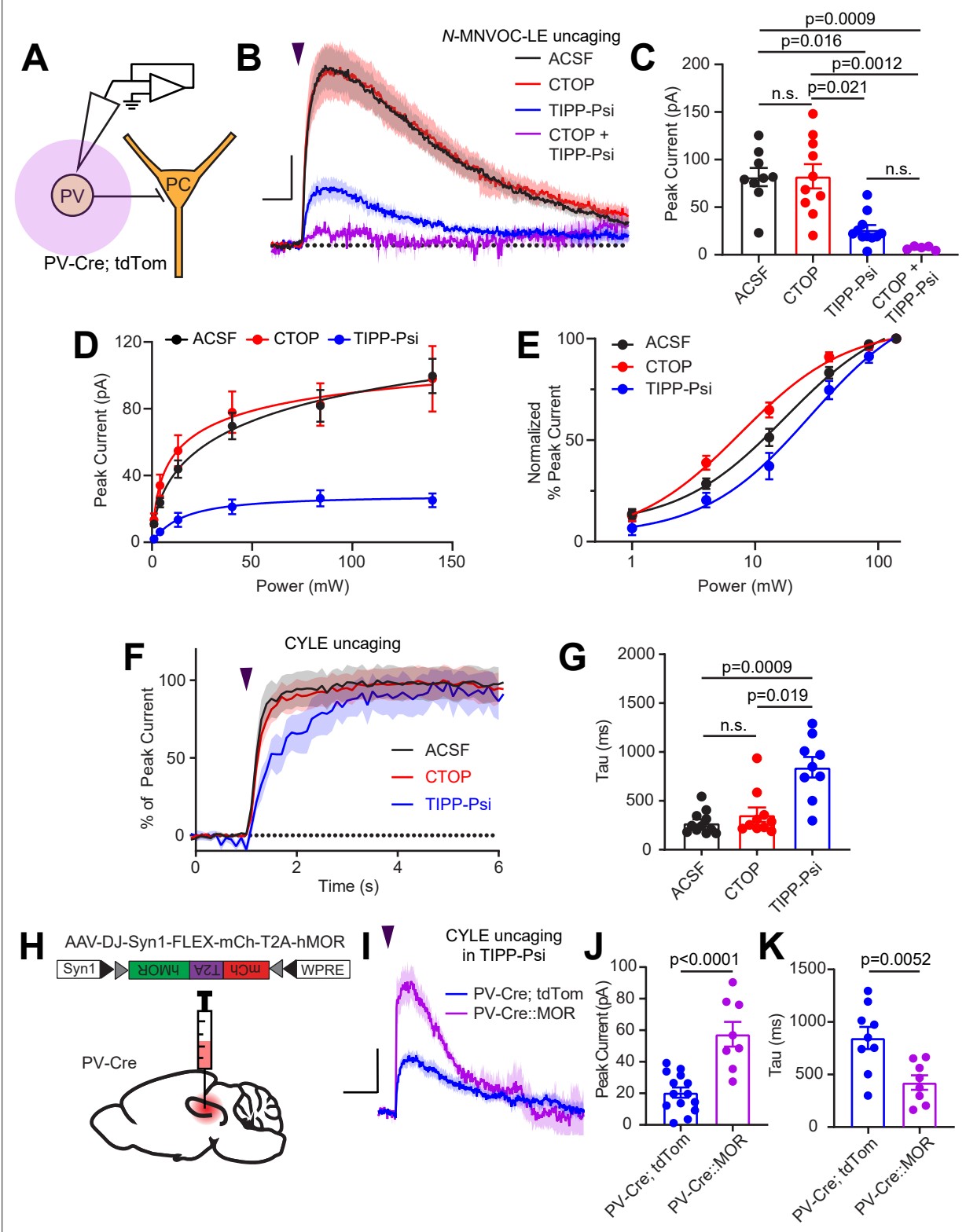

**Figure 4.** Enkephalin evokes outward currents in CA1 parvalbumin (PV) interneurons through both mu and delta opioid receptors. (**A**) Schematic of whole-cell voltage clamp recording configuration from PV interneurons with peptide uncaging. (**B**) Average outward currents evoked by photoactivation of *N*-MNVOC-LE (6 μM) with an 84 mW light flash in the absence (black, artificial cerebrospinal fluid [ACSF], n = 9 cells from five mice) and presence of mu and delta opioid receptor antagonists (red, CTOP, n = 10 cells from six mice; blue, TIPP-Psi, n = 11 cells from six mice; purple, CTOP+ TIPP-Psi, n = 5 cells from three mice). Scale bar: x = 5 s, y = 20 pA. (**C**) Summary of peak current amplitudes shown in **B**. (**D**) Linear optical power-response curve of

*Figure 4 continued on next page*

*Figure 4 continued*

peak current as a function of light intensity, in the absence (ACSF, black, n = 9 cells per laser intensity) and presence of either CTOP (red, n = 10 cells) or TIPP-Psi (blue, n = 11 cells). (**E**) Logarithmic optical power-response curves of the data in **D** normalized to the maximal peak current observed in each condition. (**F**) Rising phase of the average peak-normalized outward currents evoked by photoactivation of CYLE (6 µM) with an 84 mW light flash in the absence (black, ACSF, n = 11 cells from four mice) and presence of mu and delta opioid receptor antagonists (red, CTOP, n = 10 cells from four mice; blue, TIPP-Psi, n = 12 cells from four mice). (**G**) Time constants of current activation in response to photoactivation of CYLE from F. (**H**) Schematic of viral Cre-dependent mu opioid receptor over-expression in CA1 of PV-Cre mice. (**I**) Average outward currents evoked by photoactivation of CYLE by an 84 mW light flash in the presence of TIPP-Psi in either PV-Cre; tdTom mice (blue, data from **B**) or PV-Cre mice overexpressing the mu opioid receptor (purple, n = 8 cells from three mice). Scale bar: x = 10 s, y = 20 pA. (**J**) Summary of current amplitudes shown in I. (**K**) Time constants of current activation in response to photoactivation of CYLE.

The online version of this article includes the following source data and figure supplement(s) for figure 4:

**Source data 1.** Power-response curves and onset kinetics at somato-dendritic MOR and DOR and MOR currents after overexpression.

**Figure supplement 1.** Sensitivity of somato-dendritic currents to the G protein-coupled inward rectifier K$^+$ (GIRK) blocker Ba$^{2+}$ and mu opioid receptor expression level.

**Figure supplement 1—source data 1.** Somato-dendritic currents in Ba$^{2+}$ and ZD7288 and correlation between mCherry fluorescence and MOR currents.

To identify the ion channels underlying the MOR- and DOR-mediated outward currents, we applied the GIRK channel blocker Ba$^{2+}$ (1 mM) while delivering strong light flashes to uncage *N*-MNVOC-LE, in the absence and presence of CTOP or TIPP-Psi. Consistent with a primary role of GIRK channels, Ba$^{2+}$ blocked the majority, but notably not all, of the current mediated by both MOR and DOR to the same extent (*Figure 4—figure supplement 1A, B*) (Ba$^{2+}$ in ACSF: 67.9% ± 4.9%, n = 8 cells; Ba$^{2+}$ in CTOP: 59.6% ± 9.7%, n = 10 cells; Ba$^{2+}$ in TIPP-Psi: 67.7% ± 9.1%, n = 11 cells; no significant differences, ordinary one-way ANOVA). At DORs, inclusion of the HCN channel blocker ZD7288 (1 µM) did not further block the current, suggesting the involvement of additional ion channels (Ba$^{2+}$, ZD7288 in CTOP: 74.0% ± 5.6%, n = 9 cells; no significant difference, unpaired t-test). Due to the small size of the Ba$^{2+}$-insensitive MOR-mediated current, we did not examine the effect of ZD7288 at MOR.

One possible explanation for the slow kinetics and low efficacy of MOR-mediated GIRK activation, as well as slow kinetics of synaptic suppression, is relatively low cell surface expression of MORs in comparison to DORs. In LC, reducing available surface MORs with a covalent antagonist leads to a reduction not only in the amplitude of MOR-mediated currents, but also a slowing of activation kinetics (*Williams, 2014*). To test this hypothesis, we virally overexpressed human MOR (hMOR) with an mCherry tag in *Pvalb^Cre^* mice and probed the resulting enhanced MOR signaling with CYLE in TIPP-Psi (*Liu et al., 2021*; *Figure 4H1*). As predicted, hMOR overexpression enhanced both the magnitude (57.5 ± 7.8 pA, n = 8 cells, p < 0.0001, unpaired t-test) and the kinetics (421.8 ± 68.7 ms, n = 8 cells, p = 0.0052, unpaired t-test) of the MOR-mediated current evoked with a strong light flash in comparison to those recorded from *Pvalb^Cre^/Rosa26-lsl-tdTomato* mice (*Figure 4I–K*). Both parameters correlated strongly with mCherry fluorescence as an indicator of expression level (peak: r = 0.8314, tau on: r = –0.8538, Pearson's correlation coefficient) (*Figure 4—figure supplement 1C, D*). These results indicate that low MOR expression levels can account for the surprisingly modest effects of MOR activation in the somato-dendritic compartment of PV-BCs.

## MORs and DORs do not functionally interact in CA1 PV-BCs

The apparent co-expression of MORs and DORs in the somato-dendritic compartment is a minimal requirement for functional interactions between receptors. We therefore asked if MORs and DORs undergo heterologous desensitization such that desensitization of one receptor perturbs the function of the other. We first confirmed that prolonged exposure to DAMGO (1 µM) caused desensitization of the resulting outward current (*Figure 5A*). After incubating slices in DAMGO for at least 10 min to maximally desensitize MOR, power-response curves were obtained in the presence of DAMGO, such that subsequent photorelease of LE would only activate DORs (*Figure 5B*). We compared these responses to those evoked in naïve slices bathed in the MOR antagonist CTOP. Indicative of a lack of heterologous desensitization, neither the efficacy nor potency of LE at DORs was affected by MOR desensitization (EC$_{50}$ value of LE in the presence of DAMGO: 5.12 ± 0.38 mW, n = 9 cells; CTOP: 6.00 ± 0.42 mW, n = 7 cells) (*Figure 5C and D*). Similarly, prolonged exposure to deltorphin II (1 µM) caused desensitization of the outward current (*Figure 5E*). Desensitization of DORs using deltorphin II did not

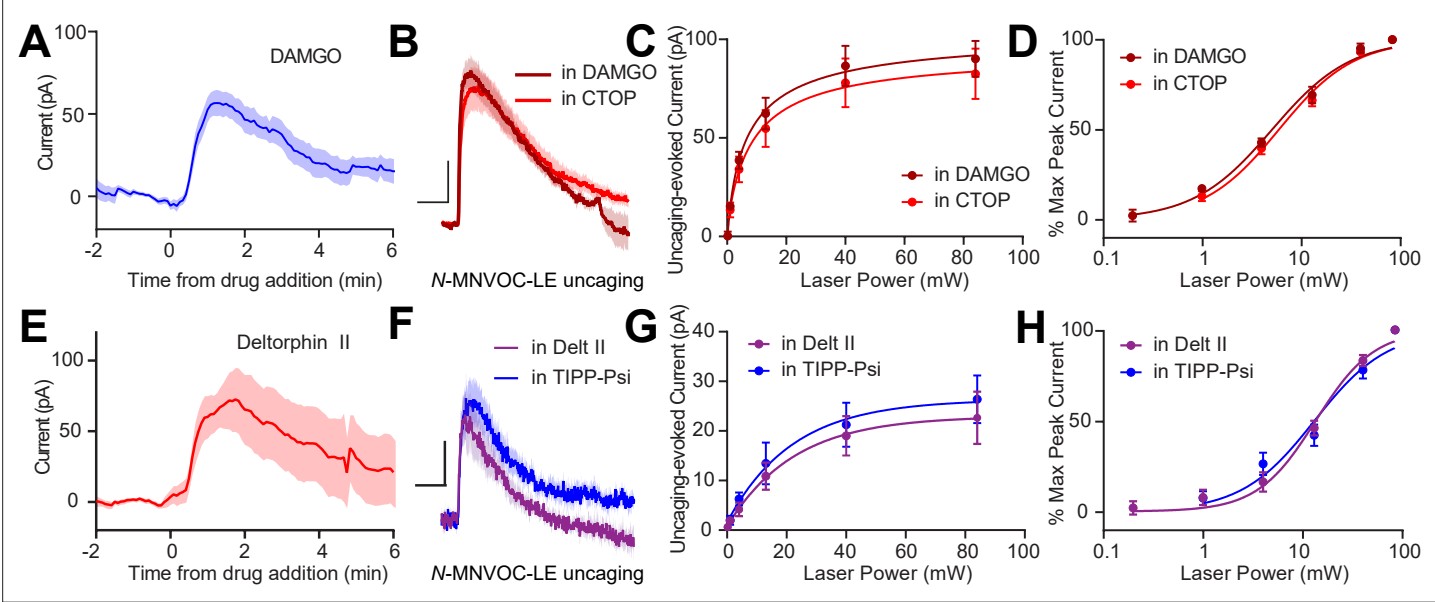

**Figure 5.** Somato-dendritic mu and delta opioid receptors do not exhibit heterologous desensitization. (**A**) Average outward current evoked by sustained bath application of DAMGO (n = 9 cells from six mice). (**B**) Average outward currents evoked by photoactivation of *N*-MNVOC-LE either in the presence of CTOP (red, data from 4B) or in the presence of DAMGO, after desensitization (brick red, n = 9 cells from four mice). Scale bars: x = 10 s, y = 25 pA. (**C**) Linear optical power-response curve of peak current as a function of light intensity, in the presence of either CTOP (red, n = 10 cells, data from 4C) or DAMGO (brick red, n = 9 cells). (**D**) Logarithmic optical power-response curves of the data in (**C**) normalized to the maximal peak current observed in each condition. (**E**) Average outward current evoked by sustained bath application of deltorphin II (n = 12 cells from six mice). (**F**) Average outward currents evoked by photoactivation of *N*-MNVOC-LE either in the presence of TIPP-Psi (blue, data from 4B) or in the presence of deltorphin II, after desensitization (purple, n = 8 cells from four mice). Scale bars: x = 10 s, y = 10 pA. (**G**) Linear optical power-response curve of peak current as a function of light intensity, in the presence of either TIPP-Psi (blue, n = 11 cells, data from 4C) or deltorphin II (purple, n = 8 cells). (**H**) Logarithmic optical power-response curves of the data in **F** normalized to the maximal peak current observed in each condition.

The online version of this article includes the following source data for figure 5:

**Source data 1.** Somato-dendritic currents from DAMGO and Deltorphin II and power-response curves of uncaging-evoked currents.

affect the ability of LE to elicit somato-dendritic outward currents compared to naïve slices bathed in the DOR antagonist TIPP-Psi (EC$_{50}$ value of LE in the presence of Delt II: 13.47 ± 1.10 mW, n = 7 cells; TIPP-Psi: 13.47 ± 1.10 mW, n = 11 cells). These results reveal that MORs and DORs do not undergo heterologous desensitization in CA1 PV-BCs.

MORs and DORs have been proposed to functionally interact through the formation of hetero-meric receptors such that a selective antagonist for one receptor enhances signaling at the other (*Gomes et al., 2004*). To directly probe for functional interactions of this type, we developed a new photoactivatable analogue of the MOR-selective agonist DAMGO, CNV-Y-DAMGO (*Ma et al., 2021*). We hypothesized that if these interactions are present, inclusion of TIPP-Psi in the bath would lead to a leftward shift in the optical power-response curves of CNV-Y-DAMGO, and possibly an increase in the response kinetics. We tested this by uncaging CNV-Y-DAMGO (1 µM) while measuring somato-dendritic currents in PV-BCs (*Figure 6A–E*) and eIPSCs in pyramidal neurons (*Figure 6F–J*). In both cases, TIPP-Psi did not alter either the kinetics of the response to DAMGO photorelease (GIRK tau on CNV-Y-DAMGO: 917.6 ± 75.7 ms, n = 11 cells; CNV-Y-DAMGO+ TIPP-Psi: 808.8 ± 46.5 ms, n = 7 cells; no significant difference, Mann-Whitney test; eIPSC tau on CNV-Y-DAMGO: 476.4 ± 36.9 ms, n = 8 cells; CNV-Y-DAMGO+ TIPP-Psi: 441.6 ± 28.1 ms, n = 7 cells; no significant difference, Mann-Whitney test) (*Figure 6C and H*), its maximal effect (*Figure 6D and I*), or its power dependence (EC$_{50}$ values for GIRKs in CNV-Y-DAMGO: 6.86 ± 0.68 mW, n = 8 cells; CNV-Y-DAMGO+ TIPP-Psi: 8.53 ± 0.64 mW, n = 7 cells; EC$_{50}$ values for eIPSCs in CNV-Y-DAMGO: 2.79 ± 0.44 mW, n = 9 cells; CNV-Y-DAMGO+ TIPP-Psi: 3.06 ± 0.38 mW, n = 9 cells) (*Figure 6E and J*). These results indicate that MORs and DORs do not interact in PV-BCs in a manner consistent with MOR/DOR heteromers. To confirm the lack of TIPP-Psi effect on DAMGO-mediated suppression of PV-BC output in a cell-specific manner, we optogenetically stimulated PV-BCs with Chronos, as in *Figure 1*, and asked if TIPP-Psi

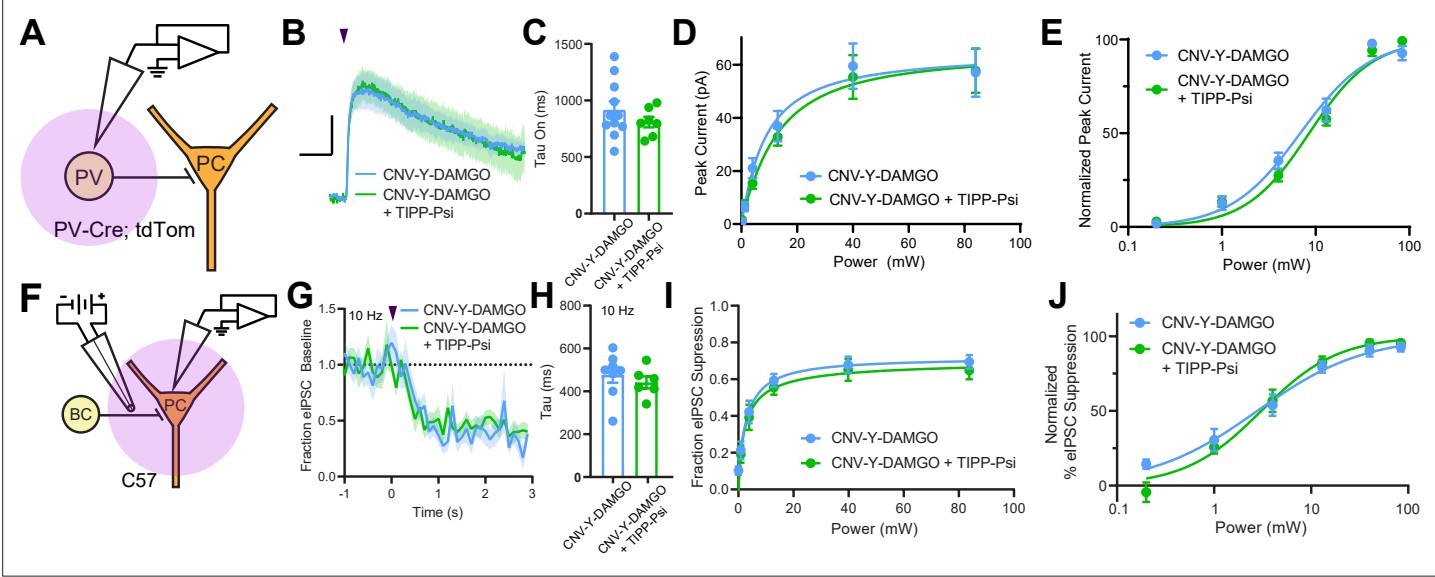

**Figure 6.** Mu and delta opioid receptors do not signal as heteromers in CA1 parvalbumin (PV) neurons. (**A**) Schematic of whole-cell voltage clamp recording configuration from PV interneurons with peptide uncaging. (**B**) Average outward currents evoked by photoactivation of CNV-Y-DAMGO with an 84 mW light flash either in the absence (sky blue, n = 8 from five mice) or presence (green, n = 7 cells from four mice) of TIPP-Psi. Scale bar: x = 10 s, y = 20 pA. (**C**) Time constants of current activation in response to photoactivation of CNV-Y-DAMGO in the absence or presence of TIPP-Psi. (**D**) Linear optical power-response curve of peak current as a function of light intensity, in the absence (sky blue) or presence (green) of TIPP-Psi. (**E**) Logarithmic optical power-response curves of the data in (**D**) normalized to the maximal peak current observed in each condition. (**F**) Schematic of the experimental configuration for photo-uncaging of opioid neuropeptides while recording electrically evoked inhibitory synaptic transmission in wild-type mice. (**G**) Average, baseline subtracted and baseline-normalized electrically evoked IPSC (eIPSC) amplitude showing the kinetics of synaptic suppression with electrical stimulation at 10 Hz in the absence (sky blue, n = 8 cells from four mice) or presence of TIPP-Psi (green, n = 8 cells from four mice). (**H**) Time constants of synaptic suppression at 10 Hz stimulation in response to photoactivation of CNV-Y-DAMGO in the absence or presence of TIPP-Psi. (**I**) Linear optical power-response curve of eIPSC suppression as a function of light intensity, in the absence (sky blue) or presence (green) of TIPP-Psi. (**J**) Logarithmic optical power-response curves of the data in **I** normalized to the maximal eIPSC suppression observed in each condition.

The online version of this article includes the following source data and figure supplement(s) for figure 6:

**Source data 1.** Power-response curves and onset kinetics of CNV-Y-DAMGO uncaging with and without TIPP-Psi.

**Figure supplement 1.** Optogenetic activation confirms that mu opioid receptor (MOR) and delta opioid receptor (DOR) do not signal as heteromers in parvalbumin (PV) terminals.

**Figure supplement 1—source data 1.** Optogenetically-evoked IPSC suppression by 300 nM DAMGO with and without TIPP-Psi.

enhanced the effect of a sub-maximal concentration of DAMGO (300 nM, *Figure 6—figure supplement 1*). Consistent with the uncaging data obtained using electrical stimulation, TIPP-Psi was again without effect (300 nM DAMGO: 0.33 ± 0.05; 300 nM DAMGO in TIPP-Psi: 0.30 ± 0.08; no significant difference, unpaired t-test).

## Discussion

### Identification of the delta opioid receptor as the primary target of enkephalin in CA1 PV-BCs

Prior models of neuromodulator actions on hippocampal interneurons have emphasized MOR expression as a primary distinctive feature of PV-BCs, as opposed to CCK-BCs (*Freund and Katona, 2007*). This results from an electrophysiological study in CA1 BCs that used the MOR agonist DAMGO to elicit outward somato-dendritic currents and suppress synaptic output (*Glickfeld et al., 2008*). Although multiple studies have demonstrated the expression of DORs, in addition to MORs, in CA1 PV neurons, the relative contributions of the two receptors to opioid modulation of CA1 PV-BCs has not been established (*Stumm et al., 2004*; *Erbs et al., 2012*; *Faget et al., 2012*). Our findings, using caged leucine-enkephalin to activate both MORs and DORs, indicate that DORs dominate cellular and synaptic responses to enkephalin, in particular at low concentrations that may be most physiologically

relevant. Notably, MOR-mediated currents of >2 pA were evoked in 22/25 cells using caged LE in TIPP-Psi, which suggests that the presence of a subpopulation of cells lacking MOR entirely do not account for the small effect. Reinforcing the dominant role of DOR, the somato-dendritic currents obtained with maximal photorelease of caged DAMGO, a full agonist of MOR G protein signaling (*Williams et al., 2013*), were also smaller than those produced by LE uncaging in CTOP (currents were apparent in 19/19 cells). Power-response curves with caged enkephalin revealed that LE activates DORs with approximately threefold greater potency than MORs in both the somato-dendritic and presynaptic compartments. Strikingly, the power-response relationships observed in the absence of antagonist closely match those obtained with MORs blocked, which underscores the dominant role of DORs in the integrated response to enkephalin. While this may reflect a greater binding affinity of LE for DORs (*Toll et al., 1998*), because somato-dendritic DOR-mediated currents are much larger than MOR-mediated currents when both receptors are saturated, this preferential recruitment of DOR signaling is also likely to result in much stronger inhibition of cellular excitability. In presynaptic terminals of PV-BCs, the strong reciprocal occlusion of synaptic suppression by saturating doses of selective MOR and DOR agonists suggests that because DOR activation by LE occurs at lower concentrations, it will occlude subsequent actions of MOR at higher doses. Given that local sources of the MOR-selective neuropeptide β-endorphin are apparently lacking in CA1 (*Bjorklund and Hokfelt, 1986*), this raises the question as to why PV-BCs express MORs at all. One possible explanation is that diurnal variation in the levels of brain-wide β-endorphin in the cerebrospinal fluid contribute to the resting excitability and tune the strength of synaptic output via PV-BC MORs, while dynamic, local release of enkephalin in CA1 produces stronger, temporally precise inhibition of cellular output through activation of DORs (*Dent et al., 1981*; *Barreca et al., 1986*).

A recent study in CA2 implicated enkephalin release from vasoactive-intestinal peptide interneurons in social memory (*Leroy et al., 2021*). This effect was attributed to DOR-mediated LTD at PV-BC synapses onto PCs (*Piskorowski and Chevaleyre, 2013*). It is currently not clear if CA2 PV-BCs also express MOR, and if their activation also drives LTD. In contrast to CA2, enkephalin-mediated presynaptic suppression of PV-BCs is reversible in CA1. Given that hippocampal DORs contribute to memory formation, and possibly, cue-related retrieval as well (*Le Merrer et al., 2011*; *Le Merrer et al., 2012*; *Le Merrer et al., 2013*; ), and that hippocampal MORs are implicated in stress-induced memory deficits, one possibility is that MOR activation in response to stress-induced β-endorphin release (*Millan et al., 1981*) occludes enkephalin actions at DOR to perturb DOR-dependent memory formation and/or retrieval. Understanding the behavioral significance of the interplay between DOR and MOR signaling will require the identification of behavior contexts that result in endogenous enkephalin release in CA1.

## Enkephalin suppresses synaptic transmission with sub-second kinetics

Although GPCRs are well established to engage effector pathways within 100 ms of exposure to agonists, data describing the kinetics of synaptic suppression by $G\alpha_{i/o}$-coupled GPCRs are sparse. A study in rat cerebellum reported rapid and transient $GABA_B$-mediated suppression of an excitatory synapse that peaked 300 ms after application of a high-frequency stimulus to drive GABA release, with detectable reduction in presynaptic $Ca^{2+}$ 100 ms after the stimulus (*Dittman and Regehr, 1997*). A similarly structured study in rat striatum observed a maximal suppression of corticostriatal transmission 500 ms after stimulating striatal neurons to release endogenous opioid neuropeptides (*Blomeley and Bracci, 2011*). Both of these studies involved relatively small quantities of neuromodulator such that rapid clearance likely obscured the intrinsic kinetics of the presynaptic signaling pathway. Here, we found that photorelease of enkephalin during high-frequency stimulation of synaptic transmission produced suppression that peaked between 1–2 s after the light flash. The high sample frequency we employed facilitated rate determination, yielding an average time constant of ~300 ms at 10 Hz. A potential caveat to our approach is that our measurements were taken from synapses that were already in a partially depressed state. Nonetheless, we observed a striking difference in the kinetics of synaptic suppression by DORs and MORs that closely matched the time constants determined for the activation of outward current in the somato-dendritic compartment. In both cases, MORs exhibited much slower kinetics (tau ~800 ms) than DORs. This was not ligand-dependent, as the same time constants were obtained using caged DAMGO (*Figure 6C and H*). This stands in contrast to prior measurements of the kinetics of GIRK activation by MORs in other cell types that found faster time

constants, similar to our measurements of DOR-mediated responses here (*Ingram et al., 1997*; *Banghart and Sabatini, 2012*; *Williams, 2014*). Interestingly, in the somato-dendritic compartment, we found that increasing MOR expression increased the MOR-evoked current activation rate. Thus, differences in MOR kinetics observed for other brain regions or cell types are likely to reflect differences in relative levels of MOR expression.

It is also notable that relatively strong activity-dependent synaptic depression due to high-frequency stimulation did not dramatically occlude synaptic suppression, indicating that release of a relatively depleted readily releasable pool of vesicles is still prone to attenuation by $G\alpha_{i/o}$-coupled GPCRs that inhibit presynaptic $Ca^{2+}$ channels. We observed a modest but significant negative correlation between the extent of synaptic suppression and the frequency of stimulation, which is consistent with voltage-dependent unbinding of $G\beta\gamma$ from VSCCs (*Bean, 1989*; *Brody et al., 1997*).

## Lack of cross-talk between MORs and DORs in CA1 PV-BCs

MORs and DORs have been suggested to physically interact via the formation of heterodimers when expressed in the same cell. Although most of the mechanistic work on MOR/DOR heteromers has been performed in cultured cells with overexpressed receptors, multiple studies have also found evidence for their occurrence in naïve brain tissue (*Gomes et al., 2004*; *Gupta et al., 2010*; *Kabli et al., 2014*; *Erbs et al., 2015*). The pharmacological framework for detecting MOR/DOR functional interactions emerges from studies in cultured cells showing that ligands for one receptor can increase the binding (in terms of $B_{max}$ but not $K_d$) and signaling efficacy of agonists for the other (*Gomes et al.,*

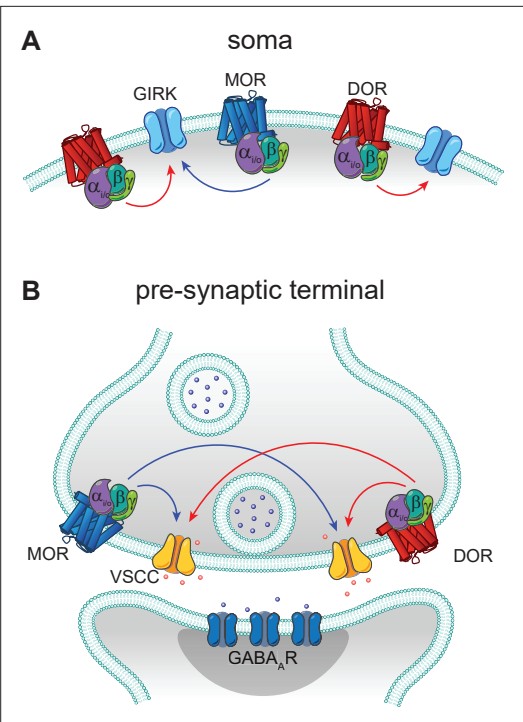

**Figure 7.** Models of mu opioid receptor (MOR) and delta opioid receptor (DOR) signaling in the soma and the pre-synaptic terminal. (**A**) In the soma, both MORs (blue) and DORs (red) signal through G protein-coupled inward rectifier K⁺ (GIRK) channels. MORs are expressed at lower levels than DORs, as the somato-dendritic currents evoked by activation of MORs alone are small and are increased by increasing MOR expression. The unidirectional occlusion observed suggests that MORs only have access to a subset of GIRKs, whereas DORs have access to a larger pool that encompasses the MOR-pool. (**B**) In the pre-synaptic terminal, MORs and DORs both act on voltage-sensitive calcium channels (VSCCs) to suppress $Ca^{2+}$ influx and inhibit vesicle release. Unlike somatic MORs and DORs, pre-synaptic MORs and DORs are bidirectionally occlusive, so that both MORs and DORs have access to the majority of VSCCs.

*2000*). Specifically, both the DOR-selective agonist deltorphin II and the selective antagonist TIPP-Psi were observed to enhance binding of DAMGO, which was accompanied by a decrease in DAMGO's $EC_{50}$ in a functional assay of MOR activation. Conversely, DAMGO, as well as the MOR antagonist CTOP, enhanced binding and reduced the $EC_{50}$ of deltorphin II. Similar enhancements of MOR activation in the presence of DOR antagonist have been observed in brain tissue using multiple functional assays of MOR signaling, including antinociceptive behavior (*Gomes et al., 2004*).

Additional evidence supporting the existence of endogenous MOR/DOR heteromers has emerged from the observation that the efficacy of bivalent MOR-DOR ligands is highly dependent on the length of the linker connecting them, which is consistent with action at a receptor complex (*Daniels et al., 2005*). Numerous studies of receptor trafficking in cultured cells indicate substantial co-localization

of MORs and DORs, as well as co-internalization upon exposure to certain agonists for one of the two receptors (e.g. *He et al., 2011*; *Derouiche et al., 2020*). In addition, biochemical studies have reported co-immunoprecipitation from naïve brain tissue using an antibody for either MORs or DORs (*Gomes et al., 2000*), or an antibody that specifically recognizes MOR/DOR heteromers (*Gupta et al., 2010*).

In contrast to these prior studies that focus on heteromers, we found no evidence for functional interactions between MORs and DORs in CA1 PV-BCs. Rather than synergistic, supralinear signaling, we observed largely parallel signaling and occlusion. If LE elicited synergistic signaling between MORs and DORs, we would predict that the power-response curve for LE with both receptors intact (control conditions) would sit to the left of the curves obtained for either receptor in isolation using selective antagonists. This was not the case. Instead, in both subcellular compartments, DOR activation accounted for the low end of the power-response curves, with MORs contributing only at higher concentrations. Strong occlusion at presynaptic terminals was observed, as simultaneous application of small molecule agonists for both receptors only slightly increased the extent of synaptic modulation in comparison to either drug alone (from 70% to ~75% suppression). Similar occlusion was also observed while monitoring presynaptic $Ca^{2+}$ transients. Interestingly, only unidirectional occlusion was observed in the somato-dendritic compartment, where MOR block had no effect on outward currents driven by high doses of LE, while DOR block dramatically reduced them. This observed sub-linear signaling suggests that DORs have access to a larger pool of GIRKs than MORs, and that GIRKs activated by MORs are completely shared between both receptor types. A model based on these results is presented in *Figure 7*.

In addition, we did not observe heterologous desensitization between MORs and DORs in the somato-dendritic compartment. In general, presynaptic inhibitory GPCRs do not desensitize (*Pennock et al., 2012*). Due to the relatively small amount of presynaptic desensitization observed (~20% with DAMGO), we did not attempt to study heterologous desensitization at presynaptic terminals. Given the strong occlusion we observed between MOR and DOR in presynaptic terminals, it remains possible that some heterologous desensitization may occur in this compartment. In opioid-naïve animals, desensitization appears to occur at the level of the receptor, likely due to C-terminus phosphorylation, rather than through the effectors (*Llorente et al., 2012*; *Leff et al., 2020*). Nonetheless, because desensitization can lead to endocytosis, and possibly conformational changes, if the receptors were physically interacting, desensitization of one receptor may be expected to impact signaling at the other.

Similarly, our findings argue against the presence of native MOR/DOR heteromers that influence cellular physiology in either the somato-dendritic or presynaptic compartments of CA1 PV-BCs, since TIPP-Psi had no effect on DAMGO potency or signaling kinetics, both of which serve as sensitive measures of receptor function. This lack of interaction between MORs and DORs is consistent with our previous observation in striatal indirect pathway neurons, wherein their actions were strictly additive, and genetic removal of either receptor neither enhanced nor suppressed the efficacy of the other (*Banghart et al., 2015*). A possible explanation is that MOR/DOR heteromers present in PV-BCs are retained in the Golgi apparatus due to a lack of Rtp4 expression (*Allen Institute for Brain Science, 2015*; *Décaillot et al., 2008*; *Saunders et al., 2018*). As this may involve sequestering MORs, it may also contribute to the surprisingly small somato-dendritic MOR-mediated GIRK currents we observed. While MOR/DOR functional interactions may be more prominent in other brain regions, our findings indicate that co-expression and co-localization in subcellular compartments do not guarantee receptor cross-talk at the cell surface.

In conclusion, DORs in CA1 PV-BCs, rather than MORs, are the primary target of the opioid neuropeptide enkephalin. Although signaling at both receptors converges on largely overlapping populations of effectors within the same subcellular compartments, MORs and DORs appear to signal predominantly in a parallel, functionally independent manner. These results imply that functional redundancy between multiple GPCRs expressed in the same neuron may be a common feature in the nervous system. Additional research is necessary to further delineate mechanisms that determine whether or not heteromers form when heterophilic receptors are present in close proximity within cells.

# Materials and methods

## Key resources table

| Reagent type (species) or resource | Designation | Source or reference | Identifiers | Additional information |
|---|---|---|---|---|
| Strain, strain background (*Mus musculus*, male and female) | C57Bl/6 | The Jackson Laboratory | Cat # 000664 RRID:IMSR_JAX:000664 | |
| Strain, strain background (*Mus musculus*, male and female) | *Pvalb^Cre* | The Jackson Laboratory | Cat # 012358 RRID:IMSR_JAX:012358 | |
| Strain, strain background (*Mus musculus*, male and female) | *Rosa26-lsl-tdTomato* (Ai14) | The Jackson Laboratory | Cat # 007914 RRID:IMSR_JAX:007914 | |
| Recombinant DNA reagent | AAV1-Syn-FLEX-Chronos-GFP | Addgene | Cat # 62722 RRID:Addgene_62722 | |
| Recombinant DNA reagent | AAVDJ-hSyn1-FLEX-mCh-T2A-FLAG-hMOR-WPRE | Banghart Lab | Addgene Plasmid #166970 | |
| Commercial assay or kit | RNAscope Fluorescent Multiplex Kit | ACD bio/Bio-Techne | Cat # 320850 | |
| Commercial assay or kit | *Pvalb* FISH probe | ACD bio/Bio-Techne | Cat # 421931-C3 | |
| Commercial assay or kit | *Oprd1* FISH probe | ACD bio/Bio-Techne | Cat # 427371-C2 | |
| Commercial assay or kit | *Oprm1* FISH probe | ACD bio/Bio-Techne | Cat # 315841 | |
| Chemical compound, drug | *N*-MNVOC-LE | **Banghart et al., 2018** | | |
| Chemical compound, drug | CYLE | Banghart Lab and NIDA Drug Supply Program **Banghart and Sabatini, 2012** | MPSP-117 (NDSP) | |
| Chemical compound, drug | CNV-Y-DAMGO | **Ma et al., 2021** | | |
| Chemical compound, drug | NBQX | HelloBio | Cat # HB0443 | |
| Chemical compound, drug | (R)-CPP | HelloBio | Cat # HB0021 | |
| Chemical compound, drug | TIPP-Psi | NIDA Drug Supply Program | MPSP-056 | |
| Chemical compound, drug | CTOP | Tocris | Cat # 1578 | |
| Chemical compound, drug | DAMGO | Tocris | Cat # 1171 | |
| Chemical compound, drug | SNC162 | Tocris | Cat # 1529 | |
| Chemical compound, drug | AlexaFluor 547 | Thermo Fisher | Cat # 10438 | |
| Chemical compound, drug | Fluo5F | Thermo Fisher | Cat # F14221 | |
| Chemical compound, drug | Picrotoxin | Sigma | Cat # P1675 | |
| Chemical compound, drug | TTX | HelloBio | Cat # HB1035 | |
| Chemical compound, drug | WIN55,212 | Tocris | Cat # 1038 | |
| Chemical compound, drug | Deltorphin II | NIDA Drug Supply Program | MPSP-036 | |
| Chemical compound, drug | ZD7288 | Tocris | Cat # 1000 | |
| Software, algorithm | MATLAB | Mathworks Inc | RRID:SCR_001622 | |
| Software, algorithm | ScanImage | **Pologruto et al., 2003** | RRID:SCR_014307 | |
| Software, algorithm | Igor Pro | WaveMetrics | RRID:SCR_000325 | |
| Software, algorithm | ImageJ | NIH | RRID:SCR_003070 | |
| Software, algorithm | Illustrator CC | Adobe Systems Inc | RRID:SCR_010279 | |
| Software, algorithm | Prism 7 | GraphPad Inc | RRID:SCR_002798 | |
| Software, algorithm | Excel | Microsoft | RRID:SCR_016137 | |

## Brain slice preparation

Animal handling protocols were approved by the UC San Diego Institutional Animal Care and Use Committee. Most experiments were conducted using postnatal day 15–32 mice of both males and females on a C57Bl/6 background. For experiments that required viral expression (*Figures 1A–E and 4H–K*, and *Figure 6—figure supplement 1*), older mice of postnatal day 25–41 (both males and females) were used. Mice were anesthetized with isoflurane and decapitated, and the brain was removed, blocked, and mounted in a VT1000S vibratome (Leica Instruments). Horizontal slices (300 μm) were prepared in ice-cold choline-ACSF containing (in mM): 25 NaHCO$_3$, 1.25 NaH$_2$PO$_4$, 2.5 KCl, 7 MgCl$_2$, 25 glucose, 0.5 CaCl$_2$, 110 choline chloride, 11.6 ascorbic acid, and 3.1 pyruvic acid, equilibrated with 95% O$_2$/5% CO$_2$. Slices were transferred to a holding chamber containing oxygenated ACSF containing (in mM): 127 NaCl, 2.5 KCl, 25 NaHCO$_3$, 1.25 NaH$_2$PO$_4$, 2 CaCl$_2$, 1 MgCl$_2$, and 10 glucose, osmolarity 290. Slices were incubated at 32°C for 30 min and then left at room temperature until recordings were performed.

## Electrophysiology

All recordings were performed within 5 hours of slice cutting in a submerged slice chamber perfused with ACSF warmed to 32°C and equilibrated with 95% O$_2$/5% CO$_2$. Whole-cell voltage clamp recordings were made with an Axopatch 700B amplifier (Axon Instruments). Data were filtered at 3 kHz, sampled at 10 kHz, and acquired using National Instruments acquisition boards and a custom version of ScanImage written in MATLAB (Mathworks). Cells were rejected if holding currents exceeded −200 pA or if the series resistance (<25 MΩ) changed during the experiment by more than 20%. For recordings measuring K$^+$ currents in PV cells (*Figure 1*), patch pipettes (open pipette resistance 2.0–3.0 MΩ) were filled with an internal solution containing (in mM): 135 KMeSO$_4$, 5 KCl, 5 HEPES, 1.1 EGTA, 4 MgATP, 0.3 Na$_2$GTP, and 10 Na$_2$phosphocreatine (pH 7.25, 286 mOsm/kg). Cells were held at −55 mV, and synaptic transmission was blocked with the addition to the ACSF of 2,3-dihydroxy-6-nitro-7-sulfamoyl-benzo(f)quinoxaline (NBQX; 10 μM), R,S-3-(2-carboxypiperazin-4-yl)propyl-1-phosphonic acid (CPP; 10 μM), picrotoxin (10 μM), and TTX (1 μM). TdTomato-expressing neurons were visualized through a Cy3 filter cube (Semrock Cy3-4040C) upon illumination with an CoolLED pE-300. For recordings measuring inhibitory synaptic transmission in mouse hippocampus, patch pipettes (2.5–3.5 MΩ) were filled with an internal solution containing (in mM): 135 CsMeSO$_3$, 10 HEPES, 1 EGTA, 3.3 QX-314 (Cl$^-$ salt), 4 Mg-ATP, 0.3 Na-GTP, and 8 Na$_2$phosphocreatine (pH 7.3, 295 mOsm/kg). Cells were held at 0 mV to produce outward currents. Excitatory transmission was blocked by the addition to the ACSF of NBQX (10 μM) and CPP (10 μM). To electrically evoke IPSCs, stimulating electrodes pulled from theta glass with ~5 μm tip diameters were placed at the border between stratum pyramidale and stratum oriens nearby the recorded cell (~50–150 μm) and two brief pulses (0.5 ms, 50–300 μA, 50 ms interval) were delivered every 20 s. The experimenters were not blinded to the pharmacological conditions employed.

## UV photolysis

Uncaging was carried out using 5 ms flashes of collimated full-field illumination with a 355 nm laser, as previously described. Light powers in the text correspond to measurements of a 10 mm diameter collimated beam at the back aperture of the objective. Beam size coming out of the objective onto the sample was 3900 μm$^2$.

## Optogenetics

AAV encoding Chronos-GFP was injected into the hippocampus of *Pvalb$^{Cre}$* pups P0-3. The virus was allowed to express for 4 weeks and then acute hippocampal slices were made as described above. For optogenetic stimulation of PV basket cell terminals, two 2 ms pulses from a blue LED (CoolLED pE-300, filtered through a 472/30 nm bandpass, Semrock [FF02-472/30-25]) were applied over the cell body of the recorded PC. The field stop of the LED was narrowed to 6600 μm$^2$ in order to limit the excitation to only the immediate axons surrounding the cell body, such that the power reaching the sample was 5–20 mW/mm$^2$.

## Two-photon calcium imaging

Two-photon imaging of axonal boutons was performed using a custom-built two-photon laser scanning microscope (*Carter and Sabatini, 2004*; *Bloodgood and Sabatini, 2007*). First, PV neurons in the CA1 region of the hippocampus were visualized using epifluorescence in a $Pvalb^{Cre}/Rosa26-lsl-tdTomato$ line and targeted recordings were made under infrared differential interference contrast (IR-DIC) on an Olympus BX51 microscope. Whole-cell current clamp recordings were made with a potassium (K)-methanesulfonate internal consisting of (in mM): 135 KMeSO$_4$, 5 KCl, 5 HEPES, 4 MgATP, 0.3 Na$_2$GTP, and 10 Na$_2$phosphocreatine. The internal also contained the Ca$^{2+}$-sensitive green fluorophore Fluo-5F (300 µM) and Ca$^{2+}$-insensitive red fluorophore Alexa Fluor-594 (30 µM). After a patch was made, the cell was allowed at least 15 min for the dye and indicator to fill the axons. Then an 800 nm laser was used to locate axonal boutons based on morphology. Once identified, line scans were made across 1–2 boutons while evoking one or five APs by injecting voltage into the cell body. Calcium transients were averaged across 30 trials, before and after drug addition. Stimulus-evoked changes in fluorescence (and the Ca$^{2+}$ signal) were reported as ΔG/Gsat, reflecting measurements of ΔG/R normalized to G/R in saturating Ca$^{2+}$ as described previously (*Bloodgood and Sabatini, 2007*).

## Data analysis

Electrophysiology data were analyzed in Igor Pro (Wavemetrics). Peak current amplitudes were calculated by averaging over a 200 ms (GIRK) or 2 ms (synaptic transmission) window around the peak. Activation time constants for GIRKs were calculated by fitting the rising phases of light-evoked currents to an exponential function. To determine magnitude of modulation by enkephalin uncaging (%IPSC suppression), the IPSC peak amplitude immediately after a flash was divided by the average peak amplitude of the three IPSCs preceding the light flash. Kinetics of synaptic modulation (*Figure 3*) were determined by averaging three stimulus trains before uncaging (at 10, 20, and 50 Hz) and fitting a bi-exponential curve to describe the synaptic depression. The curve was then divided from the stimulus train with uncaging to get the traces seen in *Figure 3B*. The time constant was then extracted from a mono-exponential fit to the suppression from the time of uncaging. The effects of drugs on IPSC suppression were calculated as the average %IPSC suppression 1–3 min after drug addition. PPR was determined by dividing Peak 2/Peak 1, where Peak 2 was calculated by subtracting the residual Peak 1 current (1 ms before second stimulus) from the absolute peak amplitude of Peak 2. Summary values are reported as mean ± SEM. Data were tested for normality using the D'Agostino and Pearson test, and the appropriate statistical tests (parametric or non-parametric) were carried out based on those results. All statistical tests were performed in GraphPad Prism except for the Skilling-Mack test, which was performed in MATLAB using code developed by Thomas Pingel [https://github.com/thomaspingel/mackskill-matlab; copy archived at swh:1:rev:8e91d5dfb95435b880ed1320727d956d2d44dd15 (*Pingel, 2016*)] . Specific statistical tests and corrections are described for each figure in the text and figure legends.

## Fluorescence in situ hybridization

Mice were deeply anesthetized with isoflurane and decapitated, and their brains were quickly removed and frozen in tissue freezing medium on dry ice. Brains were cut on a cryostat (Leica CM 1950) into 8 µm sections, adhered to SuperFrost Plus slides (VWR), and stored at –80°C. Samples were fixed with 4% paraformaldehyde, processed according to ACD RNAscope Fluorescent Multiplex Assay manual, and coverslipped with ProLong antifade reagent (Molecular Probes). Sections were imaged on a Keyence BZ-X710 Microscope at 60× magnification. The images were acquired and manually scored for the presence of fluorescent puncta and co-localization using ImageJ.

## Acknowledgements

We thank the National Institute on Drug Abuse Drug Supply Program (NDSP) for generously providing pharmacological reagents; L Sancho and E Campbell for training and assistance with two-photon microscopy; BK Lim for reagents for adenoassociated virus production; E Berg for genotyping, animal husbandry, adenoassociated virus production and administrative assistance; J Isaacson, W Birdsong, J Williams, M Lovett-Barron, and members of the Banghart Lab for helpful discussions.

## Additional information

### Funding

| Funder | Grant reference number | Author |
|---|---|---|
| National Institute on Drug Abuse | R00DA034648 | Matthew Ryan Banghart |
| National Institute of General Medical Sciences | R35GM133802 | Matthew Ryan Banghart |
| National Institute of Neurological Disorders and Stroke | U01NS113295 | Matthew Ryan Banghart |
| National Institute of Mental Health | U01NS113295 | Matthew Ryan Banghart |
| Brain and Behavior Research Foundation | NARSAD Young Investigators Award | Matthew Ryan Banghart |
| Esther A. and Joseph Klingenstein Fund | Klingenstein-Simons Fellowship in Neuroscience | Matthew Ryan Banghart |
| National Institute of General Medical Sciences | T32GM007240 | Xinyi Jenny He |
| National Institute of Neurological Disorders and Stroke | R01NS111162 | Brenda L Bloodgood |

The funders had no role in study design, data collection and interpretation, or the decision to submit the work for publication.

### Author contributions

Xinyi Jenny He, Conceptualization, Data curation, Formal analysis, Investigation, Methodology, Visualization, Writing – original draft, Writing – review and editing; Janki Patel, Connor E Weiss, Data curation, Formal analysis; Xiang Ma, Methodology, Resources; Brenda L Bloodgood, Methodology, Resources, Writing – review and editing; Matthew R Banghart, Conceptualization, Funding acquisition, Project administration, Resources, Writing – original draft, Writing – review and editing

### Author ORCIDs

Xinyi Jenny He http://orcid.org/0000-0002-3884-0596
Xiang Ma http://orcid.org/0000-0002-9164-8608
Brenda L Bloodgood http://orcid.org/0000-0002-4797-9119
Matthew R Banghart http://orcid.org/0000-0001-7248-2932

### Ethics

All procedures were performed in accordance with protocols approved by the University of California San Diego Institutional Animal Care and Use Committee (IACUC) following guidelines described in the the US National Institutes of Health Guide for Care and Use of Laboratory Animals (UCSD IACUC protocol S16171). All surgery was performed under isoflurane anesthesia.

### Decision letter and Author response

Decision letter https://doi.org/10.7554/eLife.69746.sa1
Author response https://doi.org/10.7554/eLife.69746.sa2

## Additional files

### Supplementary files
• Transparent reporting form

### Data availability

All data generated or analysed during this study are included in the manuscript and supporting files. Source data files have been provided for all figures.

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
