## [Editor Report]

This study uses novel photoactivatable opioid ligands and neurophysiological recordings in brain slices to investigate the functional interactions between the delta and mu opioid receptors in parvalbumin-expressing hippocampal interneurons. The authors demonstrate that delta and mu opioid receptors modulate potassium channels without causing heterologous desensitization, indicating that these two opioid receptor types signal independently. These findings extend previous studies by establishing the mechanisms of function of mu and delta opioid receptors in forebrain inhibitory interneurons co-expressing these receptors.

---

## [Decision Letter]

**Decision letter after peer review:**

Thank you for submitting your work entitled "Convergent, functionally independent signaling by mu and delta opioid receptors in hippocampal parvalbumin interneurons" for consideration by *eLife*. Your article has been reviewed by three peer reviewers, one of whom is a member of our Board of Reviewing Editors, and the evaluation has been overseen by Lu Chen as the Senior Editor.

Essential revisions:

Requiring experiments:

1. Regarding the experiments examining MOR-DOR functional interactions at the synaptic terminals of PV+ cells, the authors used electrical stimulation to excite PV+ cells. Electrical stimulation most likely recruits other inhibitory inputs in addition to PV+ cells, and it is clear from the literature that other types of inhibitory interneurons express MOR and/or DOR. With this electrical stimulation protocol, the authors might be recording agonist effects on convergent inputs from different MOR+ and/or DOR+ inhibitory neurons, instead of interrogating MOR-DOR cellular interactions in PV+ cells. To sustain the claim that they interrogated MOR-DOR cellular interactions specifically in PV+ inputs, the authors would need to confirm they obtain consistent findings and reach the same conclusions when stimulating specifically PV+ cells using optogenetics.

2. While the literature supports the idea that DOR is expressed by virtually all PV+ neurons, MOR expression in PV+ cells is more variable, with some PV+ cells lacking MOR expression. The authors claim that MOR has limited function in somatic excitability control, however, this limited function of MOR in soma could result, in part, from the inclusion in the analysis of recordings from MOR-negative PV+ cells. To sustain this claim, the authors would need to a. confirm MOR expression in recorded PV+ neurons, and b. test additional MOR agonists.

Without additional experiments:

3. The statistical analysis needs to be comprehensively revised, please see specific recommendations from the Reviewers.

4. The wording used by the authors to describe their experiments and findings needs to be more precise and careful; the experiments performed probe cellular functional interactions between the two receptors, not dimerization.

5. The methods need to be described more thoroughly, with the inclusion of additional details including the sex and age of the animals, blinding of experimenters, PPR experiments, new CNV-Y-DAMGO ligand, and SNC162 selectivity. Please see the specific recommendations from the Reviewers.

*Reviewer #1:*

This manuscript by Banghart et al. uses slice electrophysiology to examine the functional interactions between the delta and mu opioid receptors (DOR and MOR, respectively) in a class of neurons proposed to co-express both receptors, PV+ hippocampal interneurons. Whether these two receptors are co-expressed in neurons in intact circuits and can influence each other's functions, including through direct physical interactions and heterodimerization, is a long-standing and important question in the fields of opioid neurobiology, pain, and addiction. The studies are logically organized and generally well designed to answer this question. The experiments employ complementary and sophisticated receptor stimulation and recording paradigms, and the data presented convincingly support the authors' claim that DOR and MOR signal independently in PV+ hippocampal interneurons.

– One of the most interesting findings reported here is that MOR signaling is relatively inefficient in PV+ neurons. Supporting Figure S1 shows that not all PV+ neurons express DOR and MOR, consistent with published RNA-seq data. How was MOR expression tracked in individual PV+ recorded neurons and taken into account when interpreting the data? For example, in Figure 4C, while the authors state that "blocking MORs with CTOP had no measurable effect", it seems that there is considerable variability in the CTOP effect, with some cells showing a clear reduction in current amplitude compared to ACSF. Could it be that the cells in which no CTOP effect was observed in fact did not express MOR? This would be consistent with findings in cortical PV+ neurons, which consistently express DOR but where MOR expression is more variable (Birdsong et al. 2019 *eLife*, Smith et al. 2018 *eLife*). This clarification is important for the interpretation of many experiments. To confirm that MOR signaling is relatively inefficient in CA1 PV+ neurons, it would be helpful to test the effects of additional MOR-selective agonists on membrane potential or holding current.

– Figure 4D. Are the ACSF and CTOP peak currents statistically different? The authors indicate that "opioid-dose response curves in the presence of each antagonist revealed a larger DOR-mediated than MOR-mediated" current; however, it is unclear that this experiment revealed any MOR-mediated current at all.

– Figure 4H. What is the expression level and subcellular distribution of the hMORs, and is it comparable to that of the native MORs in cells where MOR-mediated responses are recorded? A histological analysis would be useful, if only as a control to confirm hMOR expression in PV+ cells.

– The authors indicate that they "chose SNC162 for its exceptional selectivity for DOR over MOR". It is unclear that SNC162 selectivity is superior to that of SNC80 or deltorphin II, the agonists typically used to interrogate DOR function. To justify their statement, can the authors provide published KD (or KI) data for each receptor and selectivity ratios for these three ligands?

– Can the authors provide additional information on CNV-Y-DAMGO such as its KD for MOR, selectivity for MOR vs DOR, and whether CNV-Y-DAMGO effects are lost in Oprm1 KO mice or in the presence of CTOP? At present, the utility of CNV-Y-DAMGO versus the well characterized agonist DAMGO for Figure 6 experiments is not obvious.

– The interrogation of DOR-MOR functional interactions in PV+ hippocampal neurons is unidimensional and restricted to neurophysiological effects. Ideally, one would want to clarify, or at least discuss, the function of DOR or MOR and their potential interactions in PV+ hippocampal neurons at other levels of opioid receptor biology, such as at the behavioral levels, on learning and memory.

– In the Introduction, the study by Wang et al. 2018 is described as a trafficking study. However, this study did not only use imaging of receptor trafficking to examine DOR-MOR interactions in neurons that co-express both receptors, but also electrophysiological recordings and desensitization protocols. These electrophysiological studies showed that the pharmacological elimination of DOR from the plasma membrane did not affect the ability of MOR to signal and activate GIRK channels. Additionally, Wang et al. verified MOR and DOR co-expression in recorded cells and included other levels of analysis, including behavioral studies. Since the aims of the Wang et al. 2018 study are conceptually very similar to those of the present study, and both studies conclude that MOR and DOR signal independently, it is important that these earlier findings are presented accurately to the reader in the Introduction and Discussion.

– It would also be useful to the reader to discuss the literature claiming that DOR is a dormant receptor, without function in the absence of priming stimulus, and clarify that the results presented here refute this idea.

– Please check the list of References. For example the Bean 1989 paper is cited in the Results but absent from the References section.

*Reviewer #2:*

The authors then make use of photopharmacology and calcium imaging to demonstrate that both MOR and DOR suppress presynaptic voltage gated calcium channels on PV interneurons. They demonstrate that DOR signaling is the primary driver of somato-dendritic inhibition through coupling to GIRK channels. Given that MOR and DOR were expressed on overlapping populations of neurons, they examine whether these receptors signaling as "functional heterodimers." However, they found no evidence of heterologous desensitization or functional heterodimerization, suggesting these receptors to not dimerize in hippocampal PV interneurons using the techniques they use here. It still remains rather unclear what we all mean when we say "dimers" for Class A receptors anyways. Maybe they quickly kiss one another during anterograde transport, or maybe it is packaging for internalized and degraded receptors. The authors still leave those questions very unanswered, but the study remains important in other ways and adds to our understanding of GPCR interactions. I would suggest at the outset that the authors are more careful with the term dimers throughout and clarify what they mean by the term.

Strengths:

This is a highly rigorous set of experiments, using complementary approaches to understand the functional interaction between MOR and DOR receptors in hippocampal PV interneurons. The techniques are well suited to address these questions, and the authors make use of novel photo-uncageable opioid ligands in combination with traditional pharmacological approaches to probe these interactions. Decades of research in heterologous expression systems have demonstrated that MOR and DOR can functionally dimerize, yet few studies have examined this in native systems. This study provides strong evidence for the lack of functional dimerization in PV interneurons.

Weaknesses: the central limitations of this study relation to the terminology used, the use of electrical stimulation in some cases, and the cell type selectivity.

Comments for the authors:

1) There is methodological concern with the surprising switch from optogenetic to electrical stimulation of putative PV input to pyramidal neurons. The authors mention that this is to "improve experimental throughput." However, this explanation is not sufficient given the caveats of this approach. First, the authors attempt to demonstrate that inhibitory input from CCK interneurons is not recruited by this stimulation paradigm by examining cannabinoid agonist induced depression, which they mentioned was minimal. However, 25% depression should not be dismissed, as only a subset of inhibitory interneurons in the hippocampus express CB1 (mainly CCK interneurons). This suggests that the electrical stimulation paradigm is likely also recruiting CCK input as well. While this may not in and of itself be an issue given the low expression of opioid receptors on this population, other hippocampal interneurons, such as somatostatin positive interneurons, do express opioid receptors at a high level. Therefore, the authors cannot conclude with certainty that they are only recording from PV input. There are also noticeable differences between the optical and electrical stimulation paradigms, as MOR agonist application with optical stimulation induces an increase in the PPR while there is no significant change with electrical stimulation. Lastly, the paper they site as justification for this electrical stimulation paradigm does not use bipolar electrical stimulation at any point; thus using this paper as justification is incorrect. Therefore, they should do one of three things: 1). Demonstrate these effects to do not hold for other populations of hippocampal interneurons, 2). Replicate more of their PV projection findings using optogenetic approaches, 3). Remove mention of PV and say inhibitory input instead.

2) Dimerization is a term that is frankly used for GPCRs in strange ways (think about how pentameric ion channels are described for example to reevaluate the meaning here), and so "functional heteromers" seems less accurate even. The authors need to really clarify more in the intro and discussion these distinctions and make it more clear what they mean by their terms. They've certainly provided nice data with respect to functional "interactions" between receptors, but the TIRF experiments are missing, as are critical biochemical experiments, or FRET studies to definitively rule out all interactions. There readouts limit this, as does their language about it need some refining.

*Reviewer #3:*

The question of whether different types of opioid receptors expressed in the same cell operate independently or as obligatory functional units (heterodimers) has been of great interest to the field for some time. This study tested the hypothesis that mu and delta opioid receptors (MORs and DORs respectively) that are co-expressed in hippocampal basket cells are both able to independently modulate neuronal excitability and regulate synaptic transmission. This is a potentially very exciting study as the authors conclude that these two receptors couple to similar effector systems in these cells, but do so independently. The investigators explored their hypothesis with a variety of complementary methods to assess the interactions in multiple ways. While the data largely support their conclusion, there are some concerns regarding how the data were analyzed and in some elements of the experimental design (especially related to animal age), thus suggesting caution in how to interpret their data at this point. Nonetheless, the impact of this study, if proven correct, will be great as it helps elucidate signaling interactions between different opioid receptors that respond to the same endogenous ligands and couple to similar signaling pathways.

Strengths:

1. Comparing the interaction of MORs and DORs in both the synaptic terminals and the somatodendritic compartments allows for an understanding how the cellular environment can influence interactions. The investigators used a variety of electrophysiological measures and pharmacological and genetic tools to provide a nice composite view of receptor signaling differences.

2. The use of caged opioid ligands allows for detailed assessments of the kinetics of responses.

3. The investigators explored convergent signaling, heterologous desensitization and dimerization, all important aspects of studying the different ways the receptors could interact.

Weaknesses:

1. There are concerns regarding the statistical tests performed and how the data are interpreted based upon these analyses. The methods states that all data were treated as parametric, but a justification for this is not provided, and it is not clear that this is accurate because the text reports the use of non-parametric analyses. It is apparent that the authors are interested in comparing cellular responses mediated by MORs and DORs, yet in most multiple comparison statistical tests, comparisons are only made between receptor antagonists and control and not to each other. Statistical assessment of the role of specific ion channels in the effects of MOR and DOR activation were not appropriately performed. In many places, statistical significance is ignored and results are treated as significantly different when the analyses do not support these conclusions. Therefore, some of the results should be interpreted with caution. Order effects of antagonists are considered in Figure 3, but not other figures. It was not indicated if investigator blinding to treatment was used.

2. The sex of the animals used was not reported. The age of the animals also spanned a large range (postnatal days 15-35). This is a major concern as there is considerable brain development that occurs over this age range that could produce physiological changes in the hippocampus. It is important to take age and sex into account in the interpretation of the data.

3. Heterologous desensitization was tested for somatodendritically-localized receptors, but not for synaptic receptors, given that they stated that presynaptic MORs desensitize (it was not specifically tested if DORs do). Given that there was mutual occlusion at the terminals, it is important to note whether these receptors undergo such heterologous desensitization, even if they don't heterodimerize.

4. The authors propose that DORs in the basket cells play a larger role in modulating basket cell function than MORs, and based on their data, this may be true under conditions where enkephalin predominates. Overall, the authors' conclusions are likely accurate, however it is conceivable that the results are the outcomes of testing a limited set of ligands. The authors rightly suggest that as far as physiological conditions are the focus, circulating β-endorphin could produce a different outcome. However, exogenous opioids (e.g. prescription painkillers, illicit opioids) could also differentially engage MORs and DORs and produce different outcomes due to differences in affinity or even functional selectivity.

5. Indicate the sex and ages of the animals used for each experiment.

6. Resolve discrepancy between actual statistical tests used and the statement in the methods that all data were treated as parametric. Were data tested for normality? Were outliers identified (there are a number of places where there appeared to be outliers)?

7. Dunnett's multiple comparison post hoc test is not the appropriate test for the assessment of their findings, especially as they are often making statements of comparisons between the roles of MOR and DOR. Tukey's multiple comparisons test is the more appropriate post hoc analysis. They used this test in Figure 4, but not in other figures.

8. Resolve why order effects were tested in Figure 3, but not other figures?

9. Statistical significance was treated very lightly throughout the manuscript. For example, components of Figures 1J, S1E, 2G, S2B were discussed as if they were significant (or selectively interpreted if one component was significant and another was not). For example, I am not convinced that DAMGO doesn't occlude the effect of WIN55,212-2 in S1E.

10. Figure S2B was indicated to show the role of GIRKs and HCNs in the effects of receptor activation. A one-way ANOVA was not the appropriate statistical test in this case as all this did was show that Ba2+ had no difference in its effect regardless of the drug treatment, but does not specifically test the involvement of different ion channels.

11. It was stated that DAMGO produced desensitization in Figure 1C,H. Was this statistically determined?

12. The methods do not describe the performance/analysis of PPR. How was amplitude of the second peak determined as the decay from the first hadn't completely returned to baseline?

13. The authors cite a publication in preparation as evidence of validation of their caged DAMGO peptide. This is worrisome for the interpretation of their findings without any provided validation here.

14. The authors looked at heterologous desensitization in somatodendritic signaling, but not at synaptic terminals. This is especially important as the two receptors are mutually occlusive at the terminals. Even if they are not dimers, they could produce heterologous desensitization. At the very least, the authors need to justify why they didn't explore this at synaptic terminals or present this as a limitation of their study.

15. Why do the authors think that they had less suppression as a result of uncaging than they did with bath application (Figure 1I)?

16. Some line colors in a number of figure panels are difficult to resolve due to their similarities. For example, Figure 5F-H, Figure 6.

17. Include a statement of investigator blinding.

---

## [Author Response]

Essential revisions:Requiring experiments:1. Regarding the experiments examining MOR-DOR functional interactions at the synaptic terminals of PV+ cells, the authors used electrical stimulation to excite PV+ cells. Electrical stimulation most likely recruits other inhibitory inputs in addition to PV+ cells, and it is clear from the literature that other types of inhibitory interneurons express MOR and/or DOR. With this electrical stimulation protocol, the authors might be recording agonist effects on convergent inputs from different MOR+ and/or DOR+ inhibitory neurons, instead of interrogating MOR-DOR cellular interactions in PV+ cells. To sustain the claim that they interrogated MOR-DOR cellular interactions specifically in PV+ inputs, the authors would need to confirm they obtain consistent findings and reach the same conclusions when stimulating specifically PV+ cells using optogenetics.

We agree with this statement, which pertains to ~20% of the experiments presented (Figure 2 and Figure 6G-J). Indeed, in Supporting Figure 1D-E, we show that ~25% of the eIPSC is sensitive to CB1 agonist, which suggests that our stimulation may also recruit a minor population of CB1R-expressing, presumably CCK^+^ axons. It is also likely that a small population of the recruited synapses are suppressed by both receptors, which is consistent with the occlusion observed in Figure S1E. Although we used a small bipolar stimulating electrode constructed from theta glass in order to restrict the volume of tissue stimulated to that immediately adjacent to the recorded pyramidal cell, thus enriching the eIPSC for basket-cell terminals, it is possible that we excite processes of other non-PV+ interneurons passing through the stimulated volume. We have added a statement to the Results section for Figure 1 reflecting this consideration, which is most pertinent to the potency and kinetic data presented subsequently in Figure 2.

A major reason electrical stimulation was employed was to allow the use of peptide uncaging with UV light, as the UV light also activates opsins such as Chronos. This greatly complicates analysis of uncaging responses, as one must control for the effect of a large UV light flash on subsequent oIPSCs in the absence of caged peptide – this consideration was bundled into the term “throughput” without elaboration. Because we have observed strong opsin responses to UV light flashes in the past, and subsequent alterations in blue light-evoked release, we did not attempt to combine the two optical methods in this study (the power-response experiments would be particularly impractical). We have also observed apparent opsin bleaching in response to strong UV light flashes, in particular when opsin expression is modest. We regret not stating this rationale in the manuscript. Because we introduced electrical stimulation before we introduce peptide uncaging, it felt awkward to present this justification in Figure 1, which does not involve peptide uncaging. Nonetheless we have now included it there as it is an important technical rationale that impacts the interpretation of the subsequent results.

Technical constraints aside, we have attempted to address the major finding made using electrical stimulation, namely that MOR and DOR do not functionally interact according to a receptor-heteromer model, using optogenetic stimulation of PV+ axons with Chronos in PV-Cre mice, as in Figure 1. For this experiment, we identified and applied a sub-saturating concentration of DAMGO (300 nM) in the absence and presence of the DOR antagonist TIPP-Psi (1 uM). This is very similar to the experiment shown in Figure 6I/J, but only involves a single DAMGO concentration. Because 300 nM DAMGO only produced partial suppression, if DOR antagonism enhanced MOR signaling at PV+ terminals as suggested by the MOR/DOR heteromer model, we would expect to observe greater suppression in TIPP-Psi. Consistent with our results using electrical stimulation and power-response curves with caged DAMGO, no difference was observed. This experiment is now presented in Supporting Figure S3 and has been added to the text. We think that this result strengthens our primary finding that is based solely on electrical stimulation of synaptic transmission and thank the reviewers for the suggestion.

2. While the literature supports the idea that DOR is expressed by virtually all PV+ neurons, MOR expression in PV+ cells is more variable, with some PV+ cells lacking MOR expression. The authors claim that MOR has limited function in somatic excitability control, however, this limited function of MOR in soma could result, in part, from the inclusion in the analysis of recordings from MOR-negative PV+ cells. To sustain this claim, the authors would need to a. confirm MOR expression in recorded PV+ neurons, and b. test additional MOR agonists.

a. We think that the absence of MOR in some PV neurons further supports our claim that MOR plays a limited role in somatic excitability control when considering the PV-BC cell population as a whole. When considering individual cells, our electrophysiological recordings from PV neurons do not appear to include many neurons that lack MOR responses, such that non-responders (presumably MOR-negative neurons) could account for the statistically determined difference between the average MOR and DOR-mediated somato-dendritic outward currents. A sentence was added to the discussion to address this point: “Notably, MOR-mediated currents were evoked in 22/25 cells using caged LE in TIPP-Psi, which suggests that the presence of a subpopulation of cells lacking MOR entirely do not account for the small effect.”

1. As shown Figure 4C, which reports MOR-mediated outward currents evoked by uncaging nMNV-LE, 10/11 cells had a response in the presence of TIPP-Psi. Only 1 cell was unresponsive, using a threshold of 2 pA to indicate a response.

2. As shown figure 4J, which reports the currents evoked by uncaging CYLE (PV-Cre;TdTom condition), 12/14 cells responded.

3. In Figure 6B-D, which reports the currents evoked by uncaging CNV-Y-DAMGO, although the amplitude distributions are not shown, 19/19 PV+ cells responded.

4. Considering all of these experiments, 41/44 recorded PV-BCs yielded MOR-mediated currents, suggesting that non-responders that lack MOR entirely are unlikely to account for the observation that the average LE-evoked MOR-mediated current is smaller than the average DOR-mediated current.

b. We tested two MOR agonists in this study: LE and DAMGO. DAMGO is thought to maximally engage G protein signaling and, by extension, GIRK activation. As shown in Figure 6D, currents produced by maximal uncaging of CNV-Y-DAMGO (1 uM) were ~60 pA in amplitude, which is smaller than the nMNV-LE (3 uM)-evoked currents at DOR in CTOP (Figure 4C), which averaged ~100 pA. The nMNV-LE (3 uM)-evoked currents at MOR in Tipp-Psi were ~ 25 pA in amplitude. The greater efficacy of DAMGO uncaging could be attributed to its resistance toward protease activity, in both the caged and uncaged form, which may enhance the concentration of photoreleased agonist that is able to reach the receptor and its spread compared to LE, such that DAMGO activates MORs over a larger area of dendrites than LE due to diffusion. That proteases limit peptide agonist potency and diffusion in brain slices is well established (Williams, Christie, North and Roques, J. Pharmacol Exp Ther, 1987, 243(1), 397-401; Banghart and Sabatini, Neuron, 2012, 73(2), 249-59). This may also simply reflect a greater efficacy of DAMGO than LE at MOR, although this is not well documented. Regardless, using these two different caged agonists, we observe that DOR produces larger somatodendritic currents than MOR. We added a statement to the discussion pointing out that caged-DAMGO-evoked currents at MOR were smaller than those obtained with caged LE in CTOP: “Reinforcing the dominant role of DOR, the somato-dendritic currents obtained with maximal photorelease of caged DAMGO, a full agonist of MOR G protein signaling (Williams *et al.*, 2013), were also smaller than those produced by LE uncaging in CTOP (currents were apparent 19/19 cells).”

In other work that is not included in this publication, we have measured outward currents evoked by uncaging the small molecule MOR agonist oxymorphone under ~identical conditions, and also found the responses to be quite small (~10 pA). We do not wish to include such data in this manuscript, as a separate manuscript describing that molecule in other applications is currently in preparation.

As covered in the discussion, we would also like to emphasize here that DOR exhibited much faster kinetics than MOR, both when activating somato-dendritic GIRKs, but also, clearly at least at one frequency of stimulation when suppressing synaptic transmission. MOR, instead, was profoundly slower in both assays (tau ~800 ms), when activated with either caged LE or caged DAMGO (they yielded similar time constants). Yet increasing the expression level of MOR increased both the rate of GIRK activation and peak current amplitude. Together these results strongly support our hypothesis that MOR signaling in PV-BCs is less efficient than DOR due to a lower abundance of functional receptors.

To leave room for the possibility that other MOR agonists might be more efficacious in PV-BCs than LE and DAMGO, we restricted our wording to state that DOR dominates the response to enkephalin.

Without additional experiments:3. The statistical analysis needs to be comprehensively revised, please see specific recommendations from the Reviewers.

Thank you for these constructive critiques. This feedback has improved the rigor of our study by correcting aspects of our statistical analysis. Because most, but not all, of our datasets were normally distributed, we inappropriately used primarily parametric statistics throughout, in order to run ANOVAs when assessing multiple variables. We have overhauled the statistical analysis to include the determination of normality for each dataset, and employed the appropriate parametric and non-parametric tests, with attention to the reviewers’ suggestions. A table is provided in response to Reviewer 3’s comments that summarizes the changes. The new analyses did not change the conclusions of our study or the interpretation of any single experiment. As suggested and described in detail below, we were more precise in our interpretation of several experiments (e.g. the rate determinations in Figure 2G) to more accurately reflect the statistical outcomes.

4. The wording used by the authors to describe their experiments and findings needs to be more precise and careful; the experiments performed probe cellular functional interactions between the two receptors, not dimerization.

We strongly agree with this sentiment and were very careful to not use the word “dimer” in our manuscript. In fact, we use “functional interactions” throughout with this point in mind exactly. In the introduction, beginning with the first paragraph, we very carefully described the various forms of possible functional interactions between the two receptors, which includes potential heteromerization as only one of several possibilities. A text search of the submitted manuscript did identify one mistaken use of the word “heterodimer” in a figure legend and that has been removed, as well as another intentional use in the discussion concerning models posed by other labs. When specifically discussing the heteromer hypothesis, which other groups have proposed to involve “heterodimers,” we prefer to use the word “heteromer,” as to not imply strictly dimeric interactions. In contrast, the companion manuscript by Arttamangkul et al., uses the term “heterodimer” broadly in reference to the functional interactions investigated in their study.

As described in the introduction and discussion in detail, we assessed our data in the context of multiple potential forms of functional interactions: cross-sensitization (by agonizing both receptors), cross-desensitization (heterologous desensitization), and heteromers (specifically allosteric sensitization, as evidenced with an antagonist for one receptor), all of which would manifest differently. As stated in the abstract and discussion, we concluded that our data reveal a great deal of occlusion and no evidence for cross-desensitization or sensitization, either by co-activation of both receptors, or via allosteric interactions between heteromers.

Admittedly, we did not perform biochemical or molecular imaging experiments to probe for physical interactions between MOR and DOR. Instead we used functional measures of receptor signaling-dependent cellular physiology to evaluate the proposed model for MOR/DOR heteromer signaling (among the other forms of possible functional interactions discussed). This is arguably THE most relevant measure in the context of a neuron embedded in its natural neural circuit. We added to the abstract “Thus, despite largely redundant and convergent signaling, MORs and DORs do not functionally interact in PV-BCs in a way that impacts somato-dendritic potassium currents or synaptic transmission.” to underscore this point. Also, in the introduction, we added “Thus, in naïve mice, unequivocal evidence for functional interactions between endogenous MORs and DORs in the same neurons, and in particular, for the existence of MOR/DOR heteromers that impact neuronal physiology, is lacking.

Given that we described several possibilities and *assessed* them specifically in the introduction, text, and discussion, and have added several statements clarifying that we are probing for heteromers that impact cellular physiology specifically, we hope that the language used in our manuscript is now precise and careful enough to accurately describe how our data reflect the various forms of functional interactions that may occur in neurons to impact their function.

5. The methods need to be described more thoroughly, with the inclusion of additional details including the sex and age of the animals, blinding of experimenters, PPR experiments, new CNV-Y-DAMGO ligand, and SNC162 selectivity. Please see the specific recommendations from the Reviewers.

We regret these omissions. We have added the sex and ages of animals used to the methods and indicated that experimenters were not blind to the pharmacological conditions employed. We also recalculated PPR as requested below describe the PPR calculation in more detail.

The new CNV-Y-DAMGO ligand is reported in a short pre-print that has been uploaded to BioRXiv prior to resubmission of this manuscript. A draft of this pre-print describing CNV-Y-DAMGO was, in fact, supplied to the editors with the initial submission, according to *eLife* guidelines. Perhaps it was not provided to the reviewers or simply overlooked during the initial review process – there is no mention of it in any review. Nonetheless, because preprints may be cited by *eLife* papers, assuming the reviewers accept this as sufficient validation of CNV-Y-DAMGO, this concern should be alleviated.

https://www.biorxiv.org/content/10.1101/2021.09.13.460181v1

A reference indicating the superior selectivity of SNC162 (Knapp et al.) has been added to the bibliography. https://jpet.aspetjournals.org/content/277/3/1284.long.

Reviewer #1:[…] – One of the most interesting findings reported here is that MOR signaling is relatively inefficient in PV+ neurons. Supporting Figure S1 shows that not all PV+ neurons express DOR and MOR, consistent with published RNA-seq data. How was MOR expression tracked in individual PV+ recorded neurons and taken into account when interpreting the data? For example, in Figure 4C, while the authors state that "blocking MORs with CTOP had no measurable effect", it seems that there is considerable variability in the CTOP effect, with some cells showing a clear reduction in current amplitude compared to ACSF. Could it be that the cells in which no CTOP effect was observed in fact did not express MOR? This would be consistent with findings in cortical PV+ neurons, which consistently express DOR but where MOR expression is more variable (Birdsong et al. 2019 eLife, Smith et al. 2018 eLife). This clarification is important for the interpretation of many experiments. To confirm that MOR signaling is relatively inefficient in CA1 PV+ neurons, it would be helpful to test the effects of additional MOR-selective agonists on membrane potential or holding current.

Thank you for your interest in this surprising finding. The data in Figure 4C under each pharmacological condition were obtained from different cells. We did not obtain currents before and after CTOP addition, as such antagonist flow-in experiments consume huge quantities of caged peptide. Instead we compared populations of cells recorded under each condition. We discuss the issue of MOR expression above in more depth. Frankly, we were quite surprised by the finding that CTOP did not reduce the current at all compared to control. Our interpretation, as depicted in Figure 7, is that DORs can access the same pool of GIRKs that are recruited by MORs, but not vice-versa. This might be explained, at least in part, by a relatively low abundance of MOR compared to DOR. As discussed, we also tested caged DAMGO, a full, agonist of MOR G protein signaling, and found that the maximal current evoked was still smaller than the DOR-isolating condition with LE.

– Figure 4D. Are the ACSF and CTOP peak currents statistically different? The authors indicate that "opioid-dose response curves in the presence of each antagonist revealed a larger DOR-mediated than MOR-mediated" current; however, it is unclear that this experiment revealed any MOR-mediated current at all.

The ACSF and CTOP peak currents are not statistically different. We isolated the MOR-mediated current using TIPP-Psi (plotted in blue). It peaks at about 25 pA (84 mW). The DOR-mediated current is plotted in red (CTOP). The ACSF and CTOP-mediated currents were statistically different at intermediate light powers, but we did not include this analysis in the manuscript and do not make claims about it, as we are unable easily explain it, and the effect sizes are small. Based on our interpretation that DOR can activate the same GIRKs as MOR (Figure 7 model), but not vice-versa, there may indeed be no MOR-mediated current in ACSF, as it is occluded by DOR. Yet it is revealed in TIPP-Psi.

– Figure 4H. What is the expression level and subcellular distribution of the hMORs, and is it comparable to that of the native MORs in cells where MOR-mediated responses are recorded? A histological analysis would be useful, if only as a control to confirm hMOR expression in PV+ cells.

In this experiment the goal was to overexpress MOR. It is difficult to judge the relative expression level and distribution compared to native MOR. We are highly confident of hMOR expression however, as the construct includes mCherry separated from MOR with a T2A self-cleaving peptide, such that mCherry fluorescence should scale stoichiometrically with MOR expression. As shown in Figure S2C, the peak amplitude and rise time of the caged LE-evoked current correlates well with mCherry fluorescence, which strongly suggests expression of functional receptor.

– The authors indicate that they "chose SNC162 for its exceptional selectivity for DOR over MOR". It is unclear that SNC162 selectivity is superior to that of SNC80 or deltorphin II, the agonists typically used to interrogate DOR function. To justify their statement, can the authors provide published KD (or KI) data for each receptor and selectivity ratios for these three ligands?

We cited the paper demonstrating this (Knapp et al). From their abstract:

“The most selective delta receptor ligand (SNC162) differed from SNC80 by the absence of the 3-methoxy substitution of the benzyl ring. The Ki for SNC162 at the delta receptor (0.625 nM) was over 8700-fold lower than that at the mu receptor (5500 nM), making this the most selective delta receptor ligand available.”

– Can the authors provide additional information on CNV-Y-DAMGO such as its KD for MOR, selectivity for MOR vs DOR, and whether CNV-Y-DAMGO effects are lost in Oprm1 KO mice or in the presence of CTOP? At present, the utility of CNV-Y-DAMGO versus the well characterized agonist DAMGO for Figure 6 experiments is not obvious.

A “supporting manuscript” containing many of these experiments was provided to *eLife* with our initial submission. Regretfully, it was apparently not provided to and/or read by the reviewers. We hope that this manuscript, which is now available as a pre-print, will satisfy any concerns about the validity of CNV-Y-DAMGO.

– The interrogation of DOR-MOR functional interactions in PV+ hippocampal neurons is unidimensional and restricted to neurophysiological effects. Ideally, one would want to clarify, or at least discuss, the function of DOR or MOR and their potential interactions in PV+ hippocampal neurons at other levels of opioid receptor biology, such as at the behavioral levels, on learning and memory.

Thank you for the suggestion. We have elaborated on our discussion of β-endorphin vs enkephalin signaling in hippocampus to suggest that stress-induced β-endorphin might act on MOR to occlude enkephalin actions at DOR that mediate memory formation or retrieval. While under review, a manuscript from the Siegelbaum Lab was published implicating DOR-mediated LTD of PV-BC synapses in CA2 in social memory. We have included this manuscript in the discussion as well.

– In the Introduction, the study by Wang et al. 2018 is described as a trafficking study. However, this study did not only use imaging of receptor trafficking to examine DOR-MOR interactions in neurons that co-express both receptors, but also electrophysiological recordings and desensitization protocols. These electrophysiological studies showed that the pharmacological elimination of DOR from the plasma membrane did not affect the ability of MOR to signal and activate GIRK channels. Additionally, Wang et al. verified MOR and DOR co-expression in recorded cells and included other levels of analysis, including behavioral studies. Since the aims of the Wang et al. 2018 study are conceptually very similar to those of the present study, and both studies conclude that MOR and DOR signal independently, it is important that these earlier findings are presented accurately to the reader in the Introduction and Discussion.

Thank you for pointing this out. We regret this omission and did not intend to imply that it was only a trafficking study, only that we wanted to highlight that aspect of their findings, in particular. We agree that the electrophysiology experiments employed are highly relevant and have updated the introduction to include this aspect of the study as well.

– It would also be useful to the reader to discuss the literature claiming that DOR is a dormant receptor, without function in the absence of priming stimulus, and clarify that the results presented here refute this idea.

We appreciate this point. We had considered discussing this topic but decided that it was beyond of the scope of our study. Our GIRK data strongly refute this idea in the somatodendritic compartment. However, in our synaptic transmission studies, the stimuli used to establish a baseline IPSC could, in principle, function as this priming stimulus.

– Please check the list of References. For example the Bean 1989 paper is cited in the Results but absent from the References section.

Thank you for pointing this out. We have ensured its inclusion.

Reviewer #2:The authors then make use of photopharmacology and calcium imaging to demonstrate that both MOR and DOR suppress presynaptic voltage gated calcium channels on PV interneurons. They demonstrate that DOR signaling is the primary driver of somato-dendritic inhibition through coupling to GIRK channels. Given that MOR and DOR were expressed on overlapping populations of neurons, they examine whether these receptors signaling as "functional heterodimers." However, they found no evidence of heterologous desensitization or functional heterodimerization, suggesting these receptors to not dimerize in hippocampal PV interneurons using the techniques they use here. It still remains rather unclear what we all mean when we say "dimers" for Class A receptors anyways. Maybe they quickly kiss one another during anterograde transport, or maybe it is packaging for internalized and degraded receptors. The authors still leave those questions very unanswered, but the study remains important in other ways and adds to our understanding of GPCR interactions. I would suggest at the outset that the authors are more careful with the term dimers throughout and clarify what they mean by the term.

We appreciate the overall positive assessment of our work. As discussed above and below, we did not use the term “functional heterodimers” or “dimers” in our manuscript. In a subset of experiments, we specifically intended to test a model put forth by Gomez et al. in 2004 in which DOR antagonists enhance MOR signaling (and vice versa), presumably through allosteric interactions between heteromeric receptors. Uniquely, our study examines this model using multiple measurements of cellular physiology. We did not intend to address whether or not heterodimers can exist between MOR and DOR in any form, but rather whether the synergistic model that has gained so much traction is physiologically relevant in one of the relatively rare classes of neurons in the brain that co-express both receptors.

Strengths:This is a highly rigorous set of experiments, using complementary approaches to understand the functional interaction between MOR and DOR receptors in hippocampal PV interneurons. The techniques are well suited to address these questions, and the authors make use of novel photo-uncageable opioid ligands in combination with traditional pharmacological approaches to probe these interactions. Decades of research in heterologous expression systems have demonstrated that MOR and DOR can functionally dimerize, yet few studies have examined this in native systems. This study provides strong evidence for the lack of functional dimerization in PV interneurons.Weaknesses: the central limitations of this study relation to the terminology used, the use of electrical stimulation in some cases, and the cell type selectivity.

We have done our best to clarify the terminology used and to address the limitations of using electrical stimulation, which is less cell-type selective than optogenetic stimulation.

Comments for the authors:1) There is methodological concern with the surprising switch from optogenetic to electrical stimulation of putative PV input to pyramidal neurons. The authors mention that this is to "improve experimental throughput." However, this explanation is not sufficient given the caveats of this approach. First, the authors attempt to demonstrate that inhibitory input from CCK interneurons is not recruited by this stimulation paradigm by examining cannabinoid agonist induced depression, which they mentioned was minimal. However, 25% depression should not be dismissed, as only a subset of inhibitory interneurons in the hippocampus express CB1 (mainly CCK interneurons). This suggests that the electrical stimulation paradigm is likely also recruiting CCK input as well. While this may not in and of itself be an issue given the low expression of opioid receptors on this population, other hippocampal interneurons, such as somatostatin positive interneurons, do express opioid receptors at a high level. Therefore, the authors cannot conclude with certainty that they are only recording from PV input. There are also noticeable differences between the optical and electrical stimulation paradigms, as MOR agonist application with optical stimulation induces an increase in the PPR while there is no significant change with electrical stimulation. Lastly, the paper they site as justification for this electrical stimulation paradigm does not use bipolar electrical stimulation at any point; thus using this paper as justification is incorrect. Therefore, they should do one of three things: 1). Demonstrate these effects to do not hold for other populations of hippocampal interneurons, 2). Replicate more of their PV projection findings using optogenetic approaches, 3). Remove mention of PV and say inhibitory input instead.

As described above, we agree with this general concern about the selectivity of electrical stimulation for recruiting PV terminals and have modified the text to include concerns about optical cross-talk to justify the transition to electrical stimulation. We have also chosen option 2 and replicated the lack of effect of Tipp-Psi on DAMGO-mediated synaptic suppression using optogenetic stimulation of PV-Cre axons.

The manuscript cited (Glickfeld et. al) does not refer to the electrical stimulation paradigm but the functional segregation of CB1R-sensitive regular-spiking (presumably CCK) basket cells and MOR-sensitive fast-spiking (presumably PV) basket cells. Thank you for noting this. Our writing was not clear. An additional sentence clarifying this point has been added.

2) Dimerization is a term that is frankly used for GPCRs in strange ways (think about how pentameric ion channels are described for example to reevaluate the meaning here), and so "functional heteromers" seems less accurate even. The authors need to really clarify more in the intro and discussion these distinctions and make it more clear what they mean by their terms. They've certainly provided nice data with respect to functional "interactions" between receptors, but the TIRF experiments are missing, as are critical biochemical experiments, or FRET studies to definitively rule out all interactions. There readouts limit this, as does their language about it need some refining.

A text search of our manuscript did not reveal the term “functional heteromer” or “functional heterodimer,” although to us this term might refer to a heteromer that is functionally relevant to cellular physiology. We are left to wonder if this comment was mistakenly directed at our manuscript instead of the companion paper by Attarmangkul et al.. As described above, we intentionally avoided the term “dimer” and it was only found in a single figure legend, mistakenly, in our manuscript.

Yet we would like to take this opportunity to reiterate that we think that our use of neuronal physiology as a readout is largely what distinguishes our study from others and makes our work important. As described in the manuscript, most studies into MOR/DOR heteromers involve assays with heterologous expression in order to incorporate imaging labels, or in cultured cells, or using biochemical methods that cannot reveal neurophysiological function. Thus the physiological relevance of such work has remained questionable. Our experiments assess only physiologically relevant interactions that impact somato-dendritic GIRK currents (which in turn impact cellular excitability) and synaptic transmission. While there are of course additional neuronal functions that could be studied (dendritic integration, synaptic plasticity, nuclear signaling, gene expression changes etc.), our study is one of the first to rigorously explore any aspect of neurophysiological function in the context of MOR/DOR functional interactions.

Reviewer #3:[…] 1. There are concerns regarding the statistical tests performed and how the data are interpreted based upon these analyses. The methods states that all data were treated as parametric, but a justification for this is not provided, and it is not clear that this is accurate because the text reports the use of non-parametric analyses. It is apparent that the authors are interested in comparing cellular responses mediated by MORs and DORs, yet in most multiple comparison statistical tests, comparisons are only made between receptor antagonists and control and not to each other. Statistical assessment of the role of specific ion channels in the effects of MOR and DOR activation were not appropriately performed. In many places, statistical significance is ignored and results are treated as significantly different when the analyses do not support these conclusions. Therefore, some of the results should be interpreted with caution. Order effects of antagonists are considered in Figure 3, but not other figures. It was not indicated if investigator blinding to treatment was used.

We regret not doing a better job with this important component of our study. Initially, we indeed used a mix of parametric and non-parametric statistics, but not all parametric tests were justified by tests for normality first. So our statement regarding all data being treated as parametric was incorrect. Detailed responses are provided below under Recommendations.

2. The sex of the animals used was not reported. The age of the animals also spanned a large range (postnatal days 15-35). This is a major concern as there is considerable brain development that occurs over this age range that could produce physiological changes in the hippocampus. It is important to take age and sex into account in the interpretation of the data.

We apologize for the omission. Both sexes were used equally in all experiments. We also indicated age for each experiment in the methods. Only a few experiments requiring gene expression were conducted in ~P28-P41 animals. Because PV promotor activity does not become strong until ~P18-22, older animals were required for optogenetic experiments in order to achieve adequate opsin or mCh-2A-hMOR expression. Notably, the MOR and DOR sensitivity observed in our recordings from P15-P32 animals using electrical stimulation (with caveats of course) are not obviously different from those observed using optogenetic stimulation in the older mice.

3. Heterologous desensitization was tested for somatodendritically-localized receptors, but not for synaptic receptors, given that they stated that presynaptic MORs desensitize (it was not specifically tested if DORs do). Given that there was mutual occlusion at the terminals, it is important to note whether these receptors undergo such heterologous desensitization, even if they don't heterodimerize.

Although we were interested in this point as well, we reasoned that the relatively small amount of desensitization observed would make this a very difficult experiment to interpret, as statistically detecting changes in DOR, for example, after only ~20% desensitization of MOR, could be very challenging.

4. The authors propose that DORs in the basket cells play a larger role in modulating basket cell function than MORs, and based on their data, this may be true under conditions where enkephalin predominates. Overall, the authors' conclusions are likely accurate, however it is conceivable that the results are the outcomes of testing a limited set of ligands. The authors rightly suggest that as far as physiological conditions are the focus, circulating β-endorphin could produce a different outcome. However, exogenous opioids (e.g. prescription painkillers, illicit opioids) could also differentially engage MORs and DORs and produce different outcomes due to differences in affinity or even functional selectivity.

We generally agree with this statement and note that DAMGO is considered to be a full, agonist of G protein signaling at MOR and that the DAMGO-evoked MOR-mediated currents were still smaller than the DOR-mediated currents obtained with caged LE in TIPP-Psi. Despite this, we have carefully chosen our wording to emphasize that DOR dominates the response to enkephalin, as to leave room for the possibility that other mu agonists might be more efficacious in PV-BCs than LE and DAMGO.

5. Indicate the sex and ages of the animals used for each experiment.

This information has been added to the methods.

6. Resolve discrepancy between actual statistical tests used and the statement in the methods that all data were treated as parametric. Were data tested for normality? Were outliers identified (there are a number of places where there appeared to be outliers)?

**Author response table 1. sa2table1:** 

Figure	Normal?		Results changed?
1D	Yes	One way ANOVA	No
1E new	No for BL(DAM)	Wilcoxon test, both significant	No
1I	Yes	One way ANOVA	No
1J new	No for BL(DAM)	Wilcoxon test, DAMGO p = 0.0186, SNC p = 0.058	DAMGO significant, SNC no longer significant
S1E	Yes	One way ANOVA w/ Tukey (changed from Dunnett’s)	No
S1G	Yes	Two way ANOVA	No
S1H	Yes	Paired t-test	New data
S1I	No	Skillings-Mack non-parametric test for grouped data	New data
2G	No	Kruskal-Wallis (non-parametric): only significant difference is for 20hz, between ACSF and TIPP-Psi	No
3C	No	Friedman test with Dunn’s multiple comparisons	No
3D	No	Friedman test with Dunn’s multiple comparisons	No
4C	No	Kruskal-Wallis (non-parametric) with Dunn’s multiple comparisons	No
4G	No	Kruskal-Wallis (non-parametric) with Dunn’s multiple comparisons	No
4J	Yes	Unpaired t-test	No
4K	Yes	Unpaired t-test	No
S2B	Yes	One way ANOVA, and t-test for CTOP condition only	No
6C	No	Mann-Whitney test, p = 0.4252	No
6H	No	Mann-Whitney test, p = 0.2824	No
S3B	Yes	Unpaired t-test, p = 0.7518	No

7. Dunnett's multiple comparison post hoc test is not the appropriate test for the assessment of their findings, especially as they are often making statements of comparisons between the roles of MOR and DOR. Tukey's multiple comparisons test is the more appropriate post hoc analysis. They used this test in Figure 4, but not in other figures.

All the Dunnett’s multiple comparisons post hoc tests have been replaced with either Tukey’s multiple comparisons (for parametric data) or with Kruskal-Wallis tests with Dunn’s multiple comparisons (for non-parametric data).

8. Resolve why order effects were tested in Figure 3, but not other figures?

We did not intend to imply that we were testing order effects. We did not include a specific statistical test to address the effect of the order of drug addiction. We simply tested the MOR and DOR agonists independently, and then to determine the effect of both drugs together, added the 2^nd^ drug. Rather than pooling the DAMGO/SNC162 data, we analyzed them separately. In principle, we could have done the same thing in Figure 1D and I but these experiments were not conducted as systematically to obtain the “both” condition so those data are pooled.

9. Statistical significance was treated very lightly throughout the manuscript. For example, components of Figures 1J, S1E, 2G, S2B were discussed as if they were significant (or selectively interpreted if one component was significant and another was not). For example, I am not convinced that DAMGO doesn't occlude the effect of WIN55,212-2 in S1E.

We have adjusted the text to accurately reflect the significance of the datasets.

For Figure S1E, we now state that “WIN55 in the presence of DAMGO produced only slightly more suppression than DAMGO alone, but this effect was not significant, suggesting some occlusion.”

For Figure 2G, we added a statement clarifying that the only difference found was between 20 Hz ACSF and 20 Hz TIPP-Psi and included the term “trend” to discuss the results at 10 and 50 Hz.

Figure S2B is discussed below.

10. Figure S2B was indicated to show the role of GIRKs and HCNs in the effects of receptor activation. A one-way ANOVA was not the appropriate statistical test in this case as all this did was show that Ba2+ had no difference in its effect regardless of the drug treatment, but does not specifically test the involvement of different ion channels.

Because we only have data for Ba2+ with ZD7288 in one condition (CTOP), we were unable to run a two-way ANOVA to test for the interactions between the ion channels. We decided to carry out a one-way ANOVA for the Ba2+ only conditions across ACSF, CTOP, and TIPP-Psi, and found no differences. We then carried out a t-test between the Ba2+ only and the Ba2+ with ZD7288 conditions in CTOP and also found no difference.

11. It was stated that DAMGO produced desensitization in Figure 1C,H. Was this statistically determined?

We have added Figure S1H that quantifies the degree of desensitization produced by DAMGO and SNC162 in both experiments. DAMGO produced desensitization 8-10 minutes after bath application for both optogenetic stimulation and electrical stimulation (Figure S1H) (opto: p = 0.0038, estim: p = 0.0001, paired t-test). SNC162 produced desensitization only with optogenetic stimulation, but not electrical stimulation (opto: p = 0.048, estim: p = 0.010, paired t-test). We compared these effects and found that DAMGO produces more desensitization than SNC162 using both stimulation protocols.

12. The methods do not describe the performance/analysis of PPR. How was amplitude of the second peak determined as the decay from the first hadn't completely returned to baseline?

In the initial analysis, it was simply calculated as the absolute peak amplitude. We have re-calculated the PPR after subtracting the residual Peak 1 current from Peak 2 and described this calculation in the methods.

13. The authors cite a publication in preparation as evidence of validation of their caged DAMGO peptide. This is worrisome for the interpretation of their findings without any provided validation here.

As described above, a draft of this manuscript was provided with the initial submission to *eLife*, and has now been uploaded to BioRxiv as a preprint.

14. The authors looked at heterologous desensitization in somatodendritic signaling, but not at synaptic terminals. This is especially important as the two receptors are mutually occlusive at the terminals. Even if they are not dimers, they could produce heterologous desensitization. At the very least, the authors need to justify why they didn't explore this at synaptic terminals or present this as a limitation of their study.

Presynaptic Gi/o-coupled GPCRs typically do not desensitize (Pennock, Dicken and Hentges, J. Neurosci. 2012, 32(30), 10192-200). The amount of desensitization in presynaptic receptors was relatively small (20-25%) and was most obvious only with DAMGO, so we didn’t think that we would be able to resolve changes in opioid efficacy after such partial desensitization in synaptic terminals. We have added a statement to the discussion to point out this limitation.

15. Why do the authors think that they had less suppression as a result of uncaging than they did with bath application (Figure 1I)?

We likely observed less suppression with uncaging because the photoreleased peptide is somewhat spatially restricted and does not access the entire dendritic tree, in contrast to bath application. Thus fewer receptors are activated on each cell when uncaging through a 60x objective.

16. Some line colors in a number of figure panels are difficult to resolve due to their similarities. For example, Figure 5F-H, Figure 6.

Thank you we have adjusted the colors.

17. Include a statement of investigator blinding.

The lack of blinding has been added to the methods.